# Digital aberration correction for enhanced thick tissue imaging exploiting aberration matrix and tilt-tilt correlation from the optical memory effect

ChulMin Oh [1,2], Herve Hugonnet[1,2], Moosung Lee[1,2,3,5] & YongKeun Park [1,2,4] ✉

Optical aberrations significantly impair microscopic image quality across various domains, including cell biology and histopathology diagnostics. Traditional adaptive optics techniques, such as wavefront shaping and guide star utilization, face challenges, especially in imaging biological tissues. Here, we introduce a computational adaptive optics approach tailored for optically thick samples. Utilizing the tilt-tilt correlation from the optical memory effect, our method detects phase differences in aberrations caused by small tilts in the incident waves. Experimental validation demonstrates our technique's capacity to enhance imaging of thick human tissues under substantial aberration conditions using a transmission-mode holotomography setup. Remarkably, our approach works robustly against sample movement, which is essential for enhanced imaging accuracy in critical biomedical applications.

In the journey from biological research to drug discovery and clinical diagnosis, high-resolution optical microscopy stands as a cornerstone, unveiling details that remain hidden from conventional microscopes. The resolution of these optical instruments, governed by the diffraction limit, depends on their numerical aperture (NA) and the wavelength of light used[1,2]. However, imperfections in the optical system, or aberrations, can severely undermine resolution, yielding blurred images. This blurring effect can be quantitatively described by the system's point spread function (PSF)[3], which delineates the response to a point-like source.

Aberrations pose significant challenges in computational imaging techniques such as synthetic aperture microscopy[4,5], Fourier ptychographic microscopy[6,7], and quantitative phase imaging (QPI)[8], which rely on image fusion to enhance resolution or to reconstruct 3D tomograms. The use of adaptive optics (AO), which measures and compensates for aberrations, can reduce image degradation caused by aberrations. While direct wavefront measurement of a point source within the image can determine aberrations[9–11], this typically requires

an invasive insertion of a guide star[12], which is not always feasible. An alternative, more universal approach to aberration detection and correction seeks to maximize image sharpness by leveraging the inherent fine structures within samples[13–20]. In traditional AO systems, this maximization process is achieved through iterative correction using wavefront modulation devices[21–23]. Alternatively, computational AO can perform maximization using previously measured holographic images without the use of modulation devices. Despite its intuitive appeal, the effectiveness of the sharpness-based method varies from sample to sample, and accurate extraction of higher-order aberrations is fraught with challenges, including the risk of converging to local maxima[24].

Recent breakthroughs in computational AO, particularly in reflection matrix imaging, have obviated the need for guide stars and reliance on sample sharpness[25–33]. This matrix-based AO technique relies on the correlation of single-scattering waves within the measured reflection or transmission matrix (Fig. 1b). It can handle a wider range of aberrations, including those induced by samples encountered in deep tissue imaging.

[1]Department of Physics, Korea Advanced Institute of Science and Technology, Daejeon, Republic of Korea. [2]KAIST Institute for Health Science and Technology, KAIST, Daejeon, Republic of Korea. [3]Institute for Functional Matter and Quantum Technologies, Universität Stuttgart, Stuttgart, Germany. [4]Tomocube, Inc., Daejeon, Republic of Korea. [5]Present address: Center for Integrated Quantum Science and Technology (IQST), University of Stuttgart, Stuttgart, Germany. ✉e-mail: yk.park@kaist.ac.kr

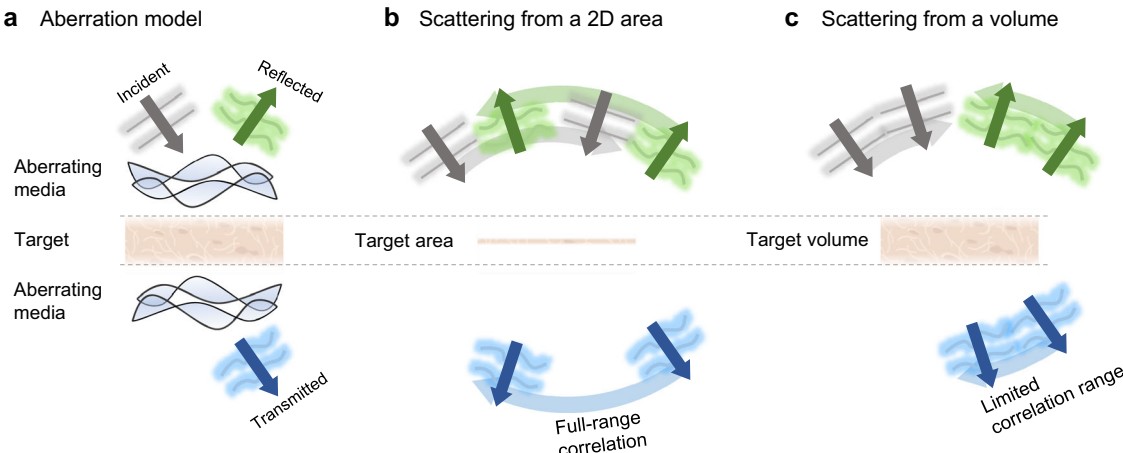

**a**  Aberration model  **b**  Scattering from a 2D area  **c**  Scattering from a volume

**Fig. 1 | Comparison of aberration models. a** Aberration model for guide-star-free adaptive optics. The region to be imaged is denoted as the target. This model simplifies the interactions of incident (gray), reflected (green), and transmitted (blue) light with the surrounding regions using the point spread functions (PSFs) of aberrating media above and below the target. **b** Conventional models assume that the target is confined to a specific plane, so that the scattered light from the target has a full-range tilt-tilt correlation. **c** In contrast, our model treats the target as a 3D object, resulting in a limited range of tilt-tilt correlation, known as the angular memory effect. Created with BioRender.com[66].

Sample-induced aberrations arise from undesired scattering outside the target plane (Fig. 1a). Since their PSF usually varies both laterally and axially, it is necessary to specify the target area before applying AO techniques[34,35]. Lateral variations in the PSF are managed by applying window functions to select an isoplanatic patch where the PSF remains constant[26,32]. Depth-dependent aberrations are addressed by collecting light reflected from the target depth using time-gating techniques derived from optical coherence tomography (OCT)[25–27,31]. However, as imaging depth increases, the prevalence of multiple scattering drastically reduces the isoplanatic patch size and undermines the depth-sectioning ability of the time gating[36]. This challenge is further compounded in transmission-mode imaging, where time gating cannot distinguish light scattered from specific depths without resorting to optical nonlinearities. Consequently, aberrations cannot be simply expressed with a single PSF, leading to unreliable convergence of the existing matrix-based AO methods.

In response to these challenges, we propose and experimentally demonstrate a computational AO method that exploits the optical memory effect. This phenomenon, characterized by field correlation against small variations in incident angle, diminishes in the presence of aberrations[37–41]. Because this correlation is preserved even in thick target objects (Fig. 1c), our approach, which restores the degraded correlation, remains effective even amidst strong aberrations and imperfect gating. The use of local correlation also ensures robust performance under sample movement by analyzing consecutive image captures. By applying severe aberrations and imaging biological tissues using a transmission-mode holotomography (3D QPI) setup, we demonstrate the potential of our method to improve the quality of deep tissue imaging in situations where existing guide-star-free AO methods fail.

## Results
### Mathematical model for aberration and scattering
To effectively detect and correct aberrations, a robust mathematical model that encapsulates both aberration and scattering phenomena is essential. To account for the sample-induced aberration, traditional AO models separately describe scattering events in aberrating media and the target plane (Fig. 1a).

Scattering in aberrating media is characterized by small scattering angles due to the large anisotropic factor of biological tissues. Due to the small scattering angle, light passing through the aberrating medium retains a considerable lateral correlation (not to be confused with angular correlation). Therefore, scattering responses within this correlation range can be described by a single PSF $P(\mathbf{r})$, and the correlated region is referred to as the isoplanatic patch (see Methods for the theoretical derivation of the correlation range). To choose a specific patch, measured waves are generally multiplied by a window function centered on that patch.

In contrast, scattering in the target plane is distinguished by a broad angular spectrum and a larger angular correlation range. Conventional AO models express target scattering by multiplying the complex reflection or transmission coefficient on the target plane $S(\mathbf{r})$. This formulation, widely employed in synthetic aperture microscopy[4,5] and Fourier ptychography[6,7], results in a full-range tilt-tilt correlation of the scattered waves (Fig. 1b). By maximizing this correlation, matrix-based AO methods, such as CLASS[25] and distortion matrix[27], can correct aberrations and obtain the 2D image of the target $S(\mathbf{r})$ (see Methods for our implementation of the existing matrix-based AO method in detail).

The overall relation between the fields (complex amplitudes) of the outgoing light $E_{\text{out}}$ and the incident light $E_{\text{in}}$ can be found by combining the two types of scattering events:

$$E_{\text{out}}(\mathbf{r}) = P_{\text{out}}(\mathbf{r}) * [(E_{\text{in}}(\mathbf{r}) * P_{\text{in}}(\mathbf{r}))S(\mathbf{r})] \tag{1}$$

where we assume $E_{\text{out}}$ and $E_{\text{in}}$ are properly windowed, $\mathbf{r}$ represents the two-dimensional coordinates on the target plane or image plane, $P_{\text{in}}$ and $P_{\text{out}}$ describe the PSFs of aberrating media in incoming and outgoing light paths, respectively, and $*$ denotes the two-dimensional convolution operator.

The use of Eq. (1) requires the target plane to be set at a specific depth inside the sample. To image deep inside biological tissues, time-gating techniques, the basic principle of OCT, have been introduced in reflection-mode microscopy to collect reflected waves at a specific depth. As forward scattering in the tissue above the target plane shows a much narrower angular spectrum than the backward scattering on the target plane, scattering outside the target plane can be incorporated into PSFs by using proper window functions.

This also highlights the limitations of the conventional guide-star-free AO methods. Strong aberration and multiple scattering deteriorate the performance of windowing and time-gating when imaging deeper. Additionally, time-gating techniques cannot provide transmitted light with depth sectioning, which limits the thickness of the target object to be on the order of a few wavelengths for transmission-mode AO. In such cases, Eq. (1) cannot be applied, and the existing guide-star-free AO methods generally converges to bad local maxima.

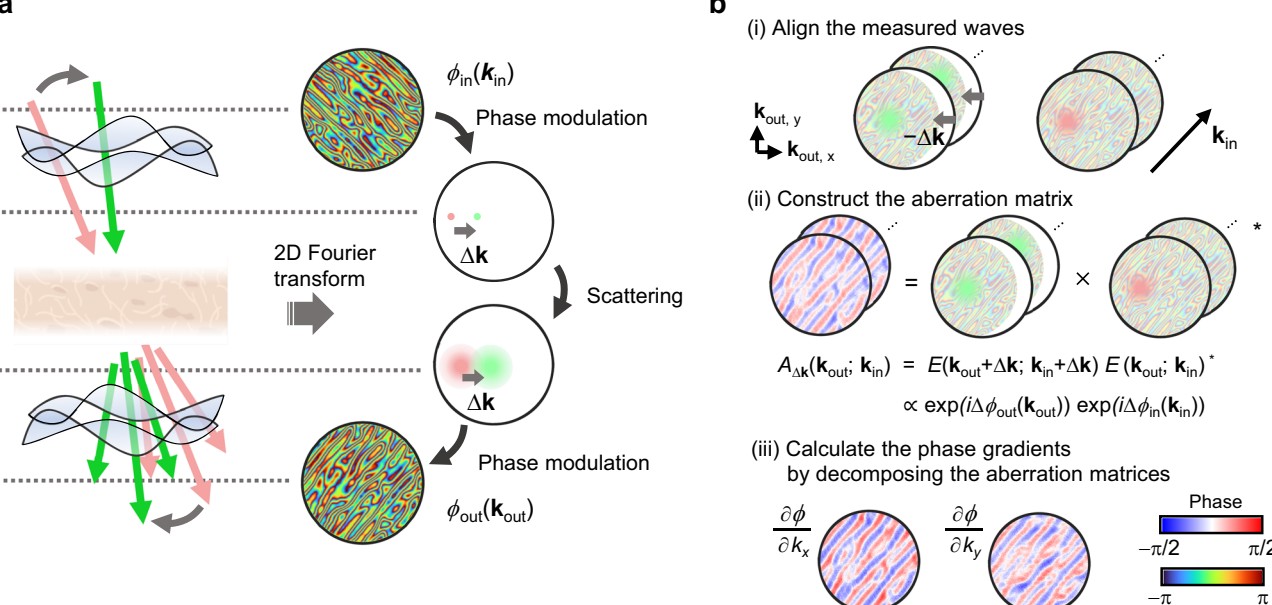

**Fig. 2 | Schematic of the aberration detection process. a** Plane waves incident on a sample are scattered and captured by varying the incident angles. According to the aberration model, the interaction of light with aberrating media can be characterized by the phase modulation in *k*-space. Scattered waves from the target volume show a shift-shift correlation in *k*-space when the tilt angle (Δ*k*) is smaller than the memory effect range. **b** The proposed method retrieves the phase gradients of the aberration functions ($\phi_{in}$, $\phi_{out}$) using the k-space images of the measured waves. These images are paired so that their incident angles fall within the memory effect range. (i) Each image pair is aligned by shifting one image according to the shift of the incident wave vector (Δ*k*, indicated by the gray arrow).

(ii) Each image is multiplied by the complex conjugate of its pair, corresponding to a column of the aberration matrix. Since the phase resulting from the target volume is canceled out, the aberration matrix manifests the outer product of the incoming and outgoing aberration functions. (iii) Phase differences of the incoming and outgoing aberration functions can be separated by decomposing the aberration matrix. Phase gradients of each aberration function can be obtained from aberration matrices of two different tilt directions ($\Delta k_1$, $\Delta k_2$). Note: the shifts shown are exaggerated for illustrative purposes and are typically less than a few pixels under experimental conditions. Figure 2a was created with BioRender.com[66].

To overcome this limitation, we generalize Eq. (1) by introducing a linear operator $T$ to represent the scattering from the target volume, rather than $S(\mathbf{r})$:

$$E_{out}(\mathbf{r}) = P_{out}(\mathbf{r}) * T[E_{in}(\mathbf{r}) * P_{in}(\mathbf{r})] \qquad (2)$$

It is worth noting that the linear operator $T$ is equivalent to the reflection or transmission matrix of a sample when imaging reflected or transmitted light, respectively[42,43]. With increased target dimension, the PSFs described by this model become averaged over the depth within the target volume.

## Detection of aberration using angular memory effect

To retrieve the PSFs $P_{in}$ and $P_{out}$ from the measured field $E_{out}$, we utilize the optical memory effect, which describes the fundamental correlation in transmitted[37–41] or reflected[37,38] optical fields under elastic scattering. While the initial paper addressed intensity correlation[40], the correlation in complex fields can be established using the Siegert relation[44]. In this work, we specifically utilize the angular memory effect, which states that if the incident wave is tilted within an angular range known as memory effect range, the scattered wave stays correlated and is likewise tilted by the same angle (Fig. 1c). By representing $T$ in matrix notation with a plane wave basis, $T(\mathbf{k}_{out}; \mathbf{k}_{in}) = \int T[e^{i\mathbf{k}_{in}\cdot\mathbf{r}}]e^{-i\mathbf{k}_{out}\cdot\mathbf{r}}d\mathbf{r}$, the angular memory effect can be described as follows:

$$\frac{|\langle T(\mathbf{k}_{out}+\Delta\mathbf{k}; \mathbf{k}_{in}+\Delta\mathbf{k})T^*(\mathbf{k}_{out}; \mathbf{k}_{in})\rangle|}{\langle |T(\mathbf{k}_{out}+\Delta\mathbf{k}; \mathbf{k}_{in}+\Delta\mathbf{k})|^2\rangle^{1/2}\langle |T(\mathbf{k}_{out}; \mathbf{k}_{in})|^2\rangle^{1/2}} = 1 - \mathcal{O}(\Delta k^2/k_{MER}^2)$$

$$(3)$$

where $\mathbf{k}_{out}$ and $\mathbf{k}_{in}$ represent the 2D wave vectors, parallel to the image plane, of outgoing and incoming light, respectively, $\Delta\mathbf{k}$ represents the shift in the wave vector corresponding to the tilt in the incident wave, $k_{MER}$ is the memory effect range, and $\mathcal{O}(\Delta k^2/k_{MER}^2)$ represents the decorrelation term on the order of $\Delta k^2/k_{MER}^2$.

To utilize Eq. (3), the outgoing fields $E_{out}$ are measured under plane wave illumination while varying the incident wave vector $\mathbf{k}_{in}$ (Fig. 2a):

$$\tilde{E}_{out}(\mathbf{k}_{out}; \mathbf{k}_{in}) = \tilde{P}_{out}(\mathbf{k}_{out})T(\mathbf{k}_{out}; \mathbf{k}_{in})\tilde{P}_{in}(\mathbf{k}_{in}) \qquad (4)$$

where we used the two-dimensional Fourier transform $\tilde{f}(\mathbf{k}) = \int f(\mathbf{r})e^{-i\mathbf{k}\cdot\mathbf{r}}d\mathbf{r}$, and the convolution theorem $\tilde{f}(\mathbf{k})\tilde{g}(\mathbf{k}) = \int f(\mathbf{r}) * g(\mathbf{r})e^{-i\mathbf{k}\cdot\mathbf{r}}d\mathbf{r}$.

If two fields are measured with slightly shifted incident wave vectors, Eq. (3) can be directly applied to Eq. (4) (Fig. 2b (i)):

$$\tilde{E}_{out}(\mathbf{k}_{out}+\Delta\mathbf{k}; \mathbf{k}_{in}+\Delta\mathbf{k})\tilde{E}_{out}^*(\mathbf{k}_{out}; \mathbf{k}_{in})$$
$$= \tilde{P}_{out}(\mathbf{k}_{out}+\Delta\mathbf{k})\tilde{P}_{out}^*(\mathbf{k})|T(\mathbf{k}_{out}; \mathbf{k}_{in})|^2\tilde{P}_{in}(\mathbf{k}_{in}+\Delta\mathbf{k})\tilde{P}_{in}^*(\mathbf{k}_{in}) + \mathcal{O}(\Delta k^2/k_{MER}^2)$$

$$(5)$$

Here, we define the aberration matrix as the left-hand side of the Eq. (5):

$$A_{\Delta\mathbf{k}}(\mathbf{k}_{out}; \mathbf{k}_{in}) = \tilde{E}_{out}(\mathbf{k}_{out}+\Delta\mathbf{k}; \mathbf{k}_{in}+\Delta\mathbf{k})\tilde{E}_{out}^*(\mathbf{k}_{out}; \mathbf{k}_{in}) \qquad (6)$$

Then, the argument of the aberration matrix manifests the aberrations for outgoing and incident light in its columns and rows,

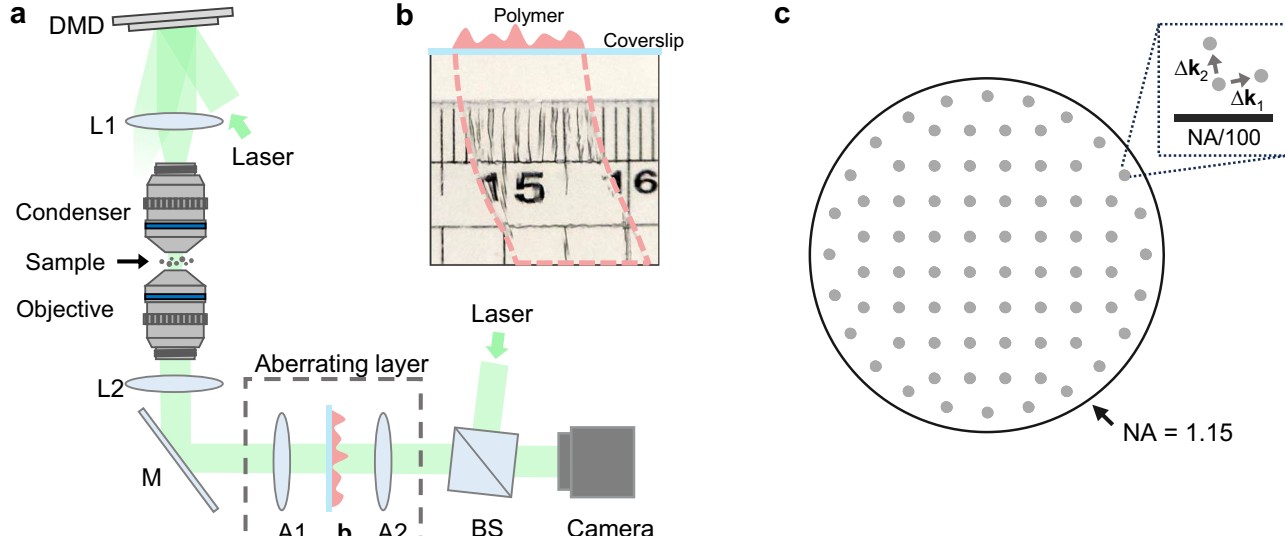

**Fig. 3 | Overview of the experimental setup. a** Our setup incorporates a transmission holotomography system. Lenses L1 and L2 establish infinity-corrected illumination and imaging paths. A1 and A2 lenses within the aberrating layer form an intermediate pupil plane, where the phase object depicted in **b** is inserted to induce controlled aberrations. Incident angles are regulated by a digital mirror device (DMD), and a beamsplitter (BS) is integrated to construct an off-axis Mach-

Zehnder interferometer, with another BS omitted here for clarity. Mirror (M) is used for beam direction. **b** A coverslip coated with polymer demonstrates the aberration effect, made evident by a distorted view of a metric ruler placed behind it. **c** The arrangement of incident wave vectors, correlating with the sine of incident angles, is composed of $N = 76$ groups. Each group contains a central angle and two additional angles tilted in different directions.

respectively (Fig. 2b (ii)):

$$\arg[A_{\Delta\mathbf{k}}(\mathbf{k}_{\text{out}};\ \mathbf{k}_{\text{in}})] = \nabla\phi_{\text{out}}(\mathbf{k}_{\text{out}}) \cdot \Delta\mathbf{k} + \nabla\phi_{\text{in}}(\mathbf{k}_{\text{in}}) \cdot \Delta\mathbf{k} + \mathcal{O}(\Delta k^2/k_{\text{MER}}^2) \tag{7}$$

where $\phi_{\text{out}}(\mathbf{k}_{\text{out}})$ and $\phi_{\text{in}}(\mathbf{k}_{\text{in}})$ represent the phases (arguments) of $\tilde{P}_{in}$ and $\tilde{P}_{out}$, which we refer to as outgoing and incoming aberration functions, respectively, and the approximation $\phi(\mathbf{k}+\Delta\mathbf{k}) - \phi(\mathbf{k}) = \nabla\phi(\mathbf{k}) \cdot \Delta\mathbf{k} + \mathbf{O}(\Delta k^2)$ has been used. The last term in Eq. (7) represents the decorrelation effect as illustrated in Fig. 1c. This term arises due to the nonzero thickness of the target volume, causing PSFs to vary within the target volume. Subsequently, the aberration functions obtained from the aberration matrix manifest averaged aberrations within the target volume.

By factorizing the aberration matrix, outgoing and incoming aberration functions can be separately obtained. The simplest way is to perform a singular value decomposition (SVD) of the aberration matrix and identify the first left and right singular vectors as the outgoing and incoming aberration functions. In this work, we use a projection power method for more accurate results (see Methods for details). For successful factorization, it is crucial to carefully select the shift of the incident wave vector ($\Delta k$) such that the magnitudes of the first and second terms on the right-hand side of Eq. (7) are greater than the last term and any noise present (see Supplementary Note 2 for detailed discussion and visualization).

As a result of factorization, the difference of aberration functions along the shift direction ($\Delta\mathbf{k}$) can be obtained. To calculate the gradients of the aberration functions, it is necessary to measure and factorize aberration matrices of at least two different shift direction ($\Delta\mathbf{k}_1$ and $\Delta\mathbf{k}_2$) (Fig. 2b (iii)). In the subsequent steps, the inverse gradient algorithm for arbitrary boundaries[45] is applied to obtain the aberration functions. Finally, the aberration is computationally corrected through the deconvolution of the outgoing fields $E_{\text{out}}$ with the retrieved outgoing PSF $P_{\text{out}}$. Here, correction of incoming aberration $P_{\text{in}}$ is not necessary because its effect is limited to constant phase offsets for each plane wave incidence. If the sample is illuminated with other than plane waves, incoming aberration must also be corrected (correction

of incoming aberration is discussed in Supplementary Note 1 for focusing inside the sample).

Ideally, the proposed method requires only three measurements to reconstruct the aberrations. However, it is necessary to measure the fields with varying incident wave vectors to ensure a uniform signal-to-noise ratio (SNR) across the entire NA. Please refer to Methods for detailed processes regarding factorization and noise filtering. Additionally, an alternative derivation using the scattering matrix can be found in Supplementary Note 3.

## Experimental setup

To validate our method, we constructed a transmission-mode off-axis holotomography setup, as illustrated in Fig. 3a. The light source is a diode-pumped solid-state laser ($\lambda = 532$ nm, SambaTM, Cobolt). The incident wave vector was controlled using a digital mirror device (DMD; V-9601, Vialux) following the method outlined in the references[46–48]. In the experiment, $N = 76$ groups of three slightly tilted wave vectors ($\Delta k/2\pi = 0.0054\lambda^{-1}$) were used to cover the NA of the objective lens (Fig. 3c). The sample was illuminated with a long working distance lens (LUMFLN60XW, NA = 1.1, Olympus), while an objective lens with a larger NA (UPLSApo60XW, NA = 1.2, Olympus) was used to collect the scattered waves. Although the NA of the objective lens is specified by the manufacturer as 1.2, our measurements indicated a slightly smaller NA, which we used in the experiment (NA = 1.15). Finally, the outgoing light interfered with a reference beam, and the holograms were captured by an image sensor (DEV-ORX-71S7M, Teledyne FLIR). By analyzing the measured holograms obtained under different illumination angles, we reconstruct the refractive index (RI) tomogram of the sample using the inverse solution of a wave equation based on the Rytov approximation[49,50]. We also present the simulation results of our method and a comparison with the existing matrix-based AO methods in time-gated reflection imaging in Supplementary Note 1.

## Aberration correction with 10-μm thick human tissue

To validate our AO method for transmission imaging of thick targets, we captured transmitted fields through human tissue as the target and a polymer-coated coverslip as the aberrating medium. The aberrating

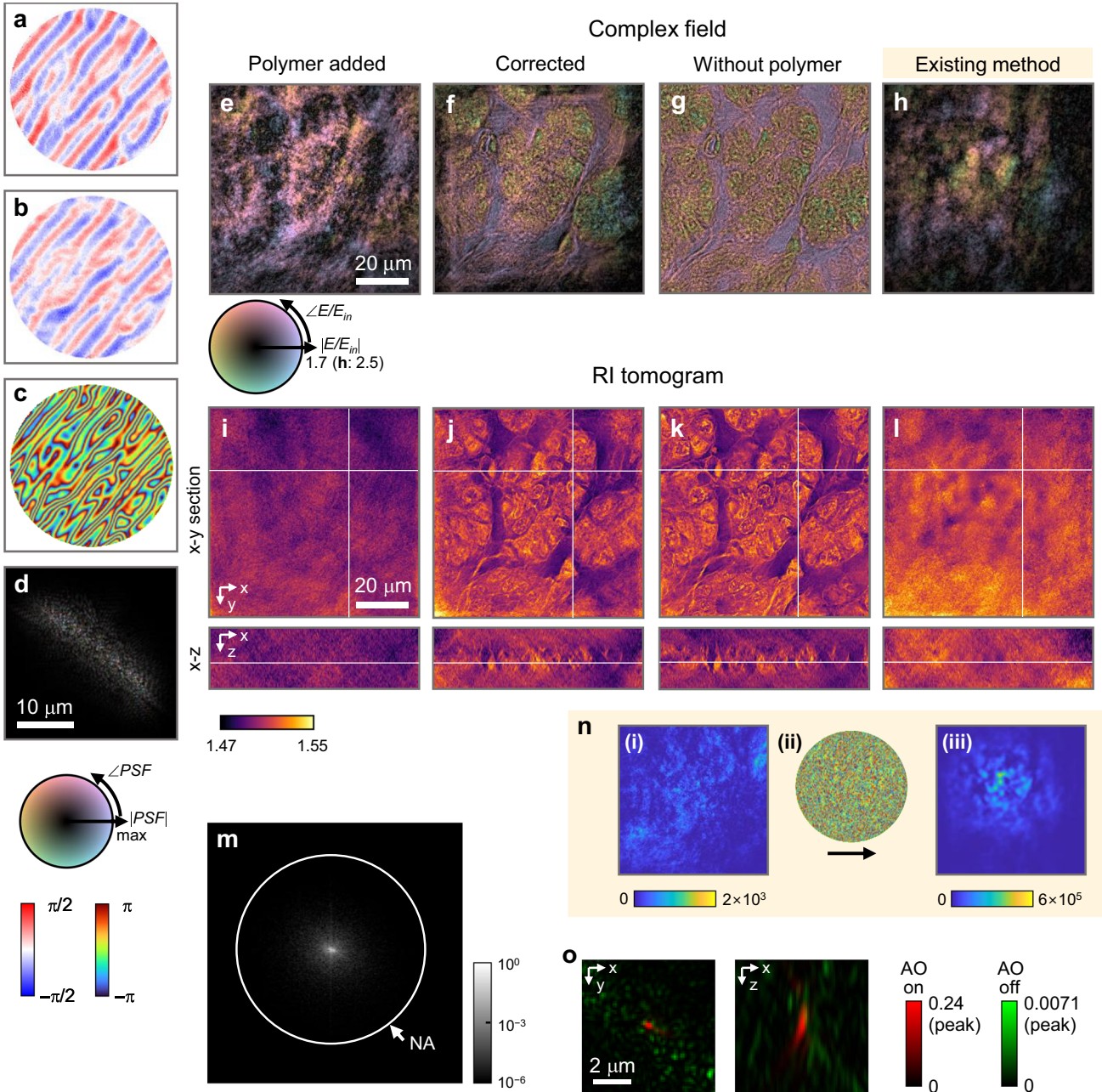

**Fig. 4 | Aberration correction with 10-µm thick human tissue. a–c** Detection of outgoing aberration functions using the proposed method: x-component (**a**) and y-component (**b**) of the phase gradient, along with the phase (**c**) of the aberration in *k*-space. **d** The point spread function (PSF) corresponds to the aberration function. **e–h** Complex fields captured under normal illumination. **i–l** Refractive index (RI) tomograms, with the z-axis aligned to the optical axis. The fields and tomograms are aberrated by the polymer (**e**, **i**), corrected using our method (**f**, **j**), acquired without the polymer (**g**, **k**), and corrected using an existing matrix-based AO method (**h**, **l**). **m** Normalized angular (Fourier) spectrum of the scattered field. **n** Application of the existing matrix-based method: 2D images $S(r)$ before (i) and after (iii) correction, and detected outgoing aberration (ii). Aberration correction results in an increase in peak intensity from $2 \times 10^3$ to $6 \times 10^5$. **o** 3D phase correlations between the aberrated fields and polymer-free fields (green), along with the correlations between the corrected fields and polymer-free fields (red).

polymer surface was created by applying an optics cleaning gel (First Contact Polymer, Photonic Cleaning Technologies) onto the coverslip, as illustrated in Fig. 3b. The polymer layer was placed between lenses A1 and A2 to initially ignore the effect of spatially varying aberration. After aberration correction, we reconstructed RI tomograms from the corrected fields using a modified Rytov approximation to improve accuracy (see Methods for details).

Figure 4 presents the outcomes of our method applied to a 10-µm thick slice of human pancreatic tissue, prepared with OpticMount™ X (RI = 1.488, BBC Biochemical) as the organic mounting medium. Initial observations of the complex fields (Fig. 4e), disturbed by the polymer

layer, failed to reveal the biological structure due to significant aberration. Similarly, the initial RI tomograms (Fig. 4i) could not delineate sample morphology.

Using our correction method, we first isolated the outgoing aberration function (Fig. 4a–c) and the corresponding PSF (Fig. 4d). Since the aberration is primarily induced by phase retardance depending on the polymer's surface profile, the retrieved aberration function also reflected the wavy texture of the polymer, as shown in Fig. 3b. We corrected the fields (Fig. 4f) by deconvolving the aberrated fields using the derived PSF and reconstructed the corresponding RI tomograms (Fig. 4j). Applying our AO method, the previously

obscured thin fibrous structures and three-dimensional features within the tissue became clear in both the fields and tomograms.

To validate the accuracy of our correction method, we measured the polymer-free fields (Fig. 4g) after removing the polymer layer and then reconstructed new RI tomograms (Fig. 4k). The structures within the corrected fields and tomograms matched those observed in their polymer-free counterparts, demonstrating the accuracy of our method. To quantify the 3D PSF of the reconstructed tomograms, we calculated 3D phase correlations between aberrated fields and polymer-free fields (see Methods for details), before and after correction (Fig. 4o). After applying our AO method, the 3D correlation becomes confined to both laterally and axially while the peak correlation is improved by a factor of 30.

Although the peak correlation is significantly enhanced, the sidelobes are not completely removed because the strong diffraction noise from the DMD (shown in Supplementary Note 5) overwhelms the narrow angular spectrum of the scattered fields (Fig. 4m) in most areas. This issue can be addressed by avoiding the use of the DMD, measuring with a larger number of incident angles, or simply increasing the tilt angle ($\Delta k$). As an example, we show that the SNR can be enhanced enough to correct weak system-induced aberrations by increasing the tilt angle ($\Delta k$) by a factor of 5, as discussed in Supplementary Note 4.

We also applied the existing matrix-based AO method discussed earlier. While the existing method could increase the intensity of the 2D images $S(\mathbf{r})$ (Fig. 4n), it failed to improve the quality of the field and tomogram through aberration correction (Fig. 4h, l). This highlights the importance of considering target thickness in transmission imaging.

## Aberration correction with 50-μm thick human tissue

To evaluate performance on thicker targets, we conducted additional experiments using a 50-μm thick human intestine tissue as the target object, prepared with Shandon Synthetic Mountant™ (RI = 1.495, Epredia). The results are summarized in Fig. 5, where we first measured the aberrated fields with the polymer introduced (Fig. 5e, left column). The aberration was then isolated using the proposed technique (Fig. 5a–d). The aberration function differed from previous results (Fig. 4a–d) because different polymer regions were used for each set of experiments. Subsequently, we obtained aberration-corrected fields (Fig. 5e, middle column) similar to the polymer-free fields (Fig. 5e, right column).

Figure 5f displays RI tomograms reconstructed from aberrated (left column), corrected (middle column), and polymer-free (right column) fields. Before applying AO, the tomograms predominantly displayed noise, obscuring any discernible structure. The application of our method, however, clearly revealed 3D tissue structures, including fibers, blood vessels, and red blood cells that were previously hidden. Specifically, section #1 revealed cells clustered in the center surrounded by fibrous material. Section #2 showed a blood vessel and red blood cells on the left, with a central fibrous matrix. Compared to the 10-μm thick tissue, the 50-μm thick tissue has a much wider angular spectrum (Fig. 5g), which overcomes the DMD noise over a larger area and improves the peak correlation after correction.

The corrected tomogram yielded nearly identical structures to the polymer-free tomogram, as evidenced by the sharp 3D correlation shown in Fig. 5h. There are, however, some differences between them because our method also corrects aberrations caused by factors other than the polymer, such as defocus and system-induced aberrations. Defocus aberration occurs when the center of the target volume is out of the focal plane. By correcting this aberration, the tomogram can be numerically refocused to the focal plane ($z = 0$). As a result, the tomogram was refocused by +3.5 μm in Fig. 5f. System-induced aberration typically arises due to imperfect optical elements and misalignment of the elements and samples. Correcting for this aberration

further improves the polymer-free tomogram, as can be observed in the corrected tomogram, where red blood cells (indicated by white arrows) show enhanced axial resolution compared to the polymer-free tomogram (more detailed analysis can be found in Supplementary Note 6).

## Memory effect in 100-μm thick human tissue and its influence on aberration correction

To assess the efficacy and boundaries of our method, we conducted an experiment involving a 100-μm thick human pancreatic tissue sample, prepared with Shandon Synthetic Mountant™ (RI = 1.495, Epredia). Due to the substantial thickness of the sample, the observed fields exhibited speckle-like patterns in both phase and intensity, as shown in Fig. 6a. The angular spectrum of this field (Fig. 6b) shows a transition beyond the single-scattering regime, with scattered fields pervading across the NA and reducing the unscattered fields compared to the spectra of the 10 and 50-μm thick tissues. In this regime, characterization of the depth-dependent aberration is critical to imaging the sample. However, this is difficult to achieve with a transmission-mode microscope due to the limited depth-sectioning mechanism, resulting in the detection of aberrations averaged over the target volume (see Supplementary Note 1 for the application of our method to depth-dependent aberrations using time-gating techniques). This limitation is depicted in Fig. 6c, where the 3D correlation of the corrected fields shows lateral confinement along with axial elongation due to the depth-dependent nature of aberrations.

Given these constraints, our focus shifted towards understanding how aberrations affect field correlations, rather than directly reconstructing the tomogram. By applying Parseval's theorem, we can express the field correlation ($g^{(1)}$ correlation) in terms of wave vectors, offering a quantifiable measure of the impact of aberrations on light transport properties:

$$C(\Delta k) = \frac{\langle \tilde{E}^{*}(\mathbf{k}_{\text{out}}; \mathbf{k}_{\text{in}}) \tilde{E}(\mathbf{k}_{\text{out}} + \Delta k; \mathbf{k}_{\text{in}} + \Delta k) \rangle_{\mathbf{k}}}{\langle |\tilde{E}(\mathbf{k}_{\text{out}}; \mathbf{k}_{\text{in}})|^2 \rangle_{\mathbf{k}}^{1/2} \langle |\tilde{E}(\mathbf{k}_{\text{out}} + \Delta k; \mathbf{k}_{\text{in}} + \Delta k)|^2 \rangle_{\mathbf{k}}^{1/2}} \quad (8)$$

where $\langle \cdot \rangle_{\mathbf{k}}$ denotes the average over the $k$-space.

Figure 6d depicts the correlations for polymer-free fields, fields measured with the polymer layer, and fields corrected by our method. Due to the thickness of the sample, these correlations exhibit an exponential decrease. The memory effect range, defined by the 1/e correlation threshold, is quantified as a shift of 0.024 $\lambda^{-1}$ in the incident spatial frequency $\Delta u = \Delta k / 2\pi$, corresponding to an angular tilt of 0.9° for normal incidence. Introducing aberration significantly accelerates the decay of the correlation compared to the polymer-free fields. Remarkably, the correlation nearly returns to its original level after aberration correction, as indicated by the gray arrow in Fig. 6d.

## Aberration correction with moving samples

Our aberration correction method is well suited for samples in motion because the aberrations are derived from the relative phase of consecutive images. Therefore, motion has little effect on the performance if the decorrelation time of the scattered fields is longer than the time interval between the consecutive captures. In terms of velocity, our method is effective for samples that move less than the diffraction limit during the interval. For example, our setup allows for movement up to 30 μm/s given the 60 Hz frame rate of the camera. Aside from movement within the target volume, the aberrations should remain unchanged throughout the total measurement. Nevertheless, we expect the time scale of aberration changes to be much larger than that of the target, since the isoplanatic patch is typically larger than the diffraction limit.

To validate the applicability of our method to dynamic samples, we observed 2-μm-diameter silica beads undergoing Brownian motion

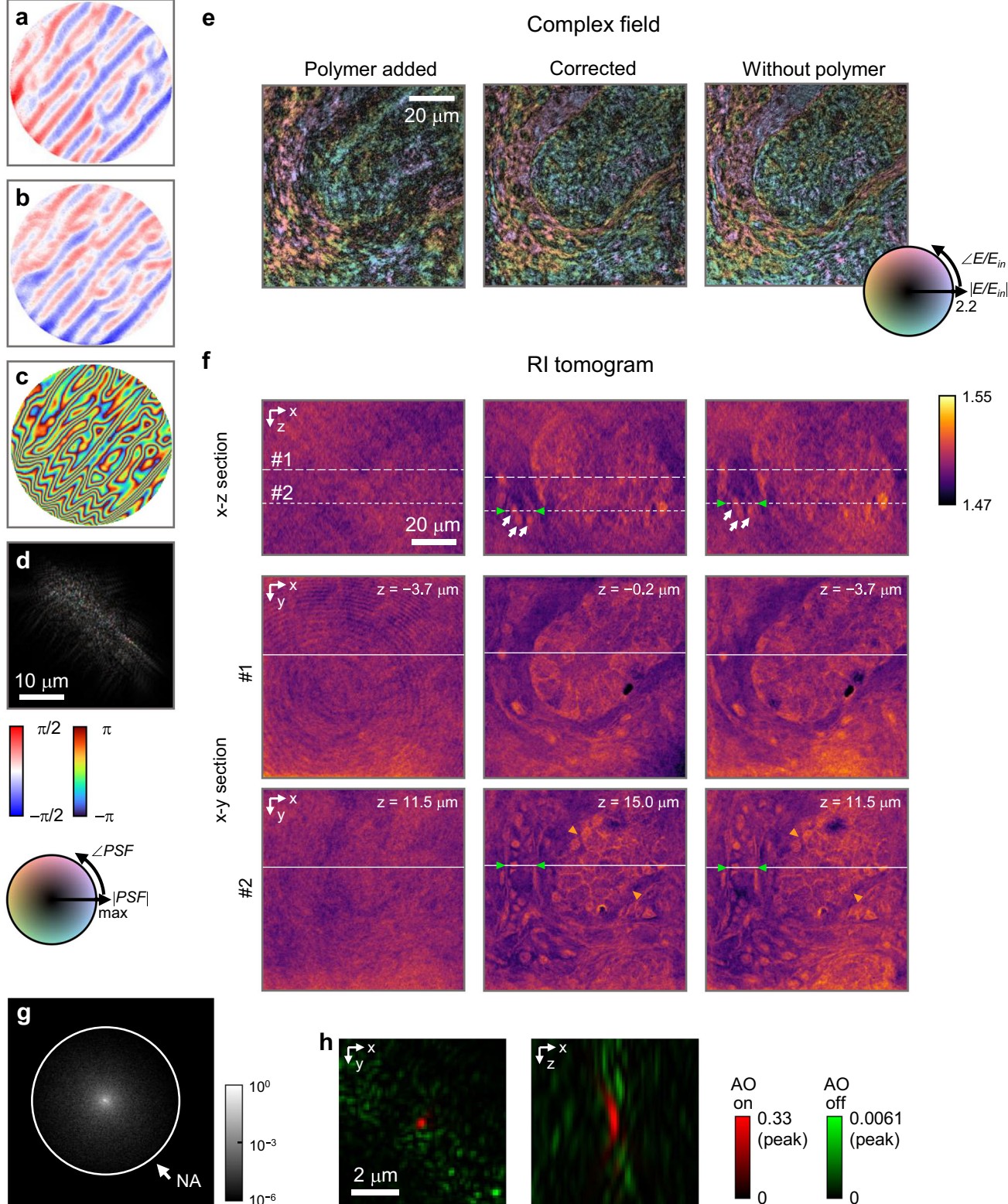

**Fig. 5 | Aberration correction with 50-µm thick human tissue. a–c** Detection of outgoing aberration functions using the proposed method: x-component (**a**) and y-component (**b**) of the phase gradient, along with the phase (**c**) of the aberration in *k*-space. **d** The point spread function (PSF) corresponding to the aberration function. **e** Complex fields under normal illumination. **f** RI tomograms. The fields and tomograms are aberrated by the polymer (left column), corrected using our method (middle column), and acquired without the polymer (right column). White arrows highlight red blood cells within a blood vessel annotated by green triangles. Orange triangles indicate a cell cluster within a fibrous matrix. The white dotted box indicates the area where the RI variance is calculated. **g** Normalized angular (Fourier) spectrum of the scattered field. **h** 3D phase correlations between the aberrated fields and polymer-free fields (green), along with the correlations between the corrected fields and polymer-free fields (red).

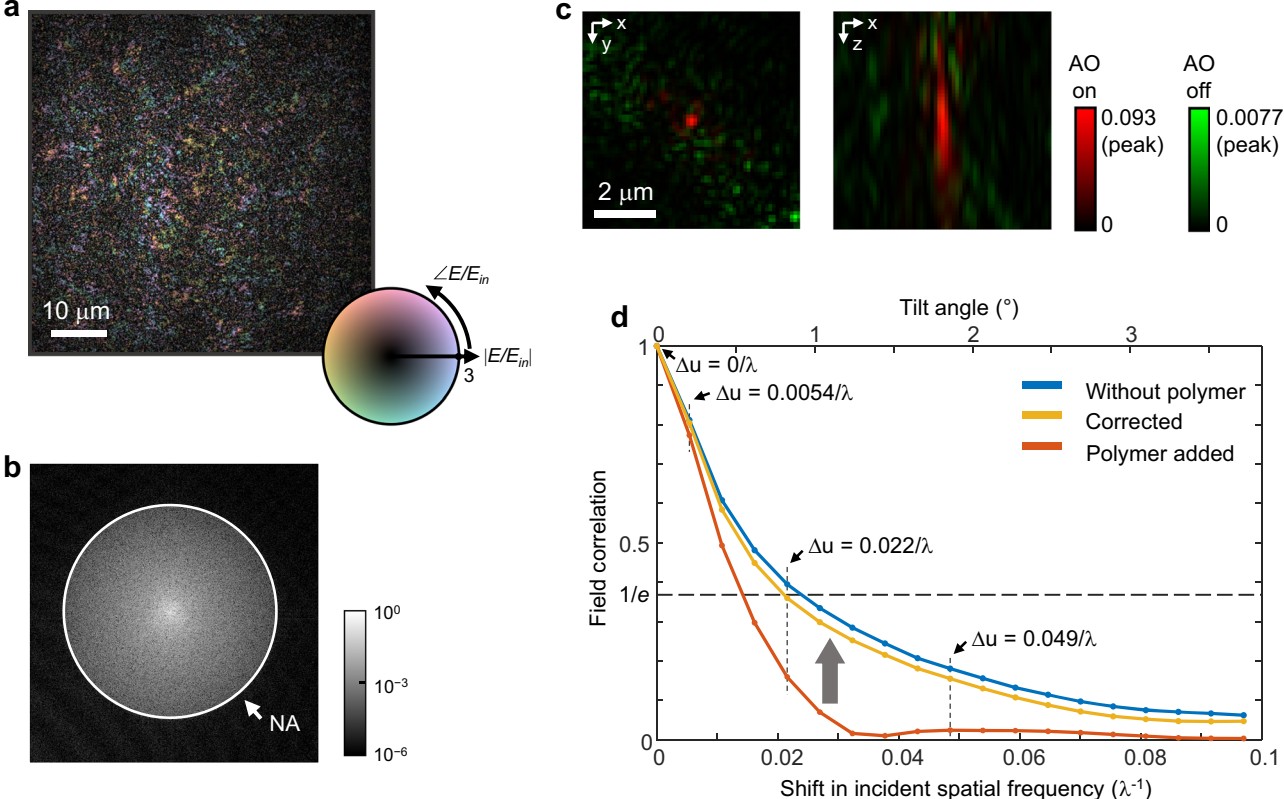

**Fig. 6 | Analyzing memory effect in 100-μm thick human tissue. a** Scattered field from the tissue under normal incidence with its Fourier spectrum displayed in the inset, revealing the speckle pattern indicative of multiple scattering. **b** Normalized angular (Fourier) spectrum of the scattered field. **c** 3D phase correlations between the aberrated fields and polymer-free fields (green), along with the correlations between the corrected fields and polymer-free fields (red). **d** A quantitative comparison of field correlations ($|C(\Delta k)|$) showcases the impact of aberration: polymer-free fields are depicted in blue, fields with polymer-induced aberration in orange, and aberration-corrected fields in yellow. The gray arrow indicates the restoration of correlation following aberration correction. Source data are provided as a Source Data file.

in water, with a field decorrelation time of 0.065 seconds. Figure 7a presents fields scattered by the moving beads and aberrated by the polymer layer. These fields do not show the shape or motion of the beads due to significant aberrations.

After correcting the aberration using our method (Fig. 7b), the fields accurately represent the shape of the beads, facilitating analysis of the bead dynamics (Fig. 7c). Notably, while all beads are inherently the same shape, their appearance varies with distance from the focal plane due to diffraction. This variation allows axial motion to be observed alongside lateral displacements. For example, one bead marked with triangles in Fig. 7c shows axial displacement, while two beads marked with arrows show independent changes in their relative positions.

**Aberration correction for spatially varying aberrations**

Aberrating media typically have a limited size of isoplanatic patch, causing spatially varying PSFs. Unlike the uniform aberrations we induced in the previous demonstration, correcting sample-induced aberrations requires selecting specific patches by properly windowing the incoming and outgoing waves. Windowing is particularly important when dealing with sample-induced aberrations, where iso-planatic patch sizes are typically much smaller than the field of view (FOV).

To demonstrate this, we sandwiched a 10-μm thick tissue between two pieces of plastic (plastic mold in BX0110, Thorlabs), as shown in Fig. 8a. For this experiment, we replaced the objective lens with a long working distance lens (LUMFLN60XW, NA = 1.1, Olympus). Since the scattering inside the plastic pieces is more anisotropic than inside the

tissue, the plastic pieces and tissue can be distinguished as aberrating media and target volume, respectively, after windowing the fields. To numerically apply the window function (Fig. 8b) to the incident fields, we scanned the incident wave vectors near the measurement points shown in Fig. 3c and summed the transmitted fields with appropriate weights to position the window at the desired location (see Methods for details). In contrast, transmitted waves can be directly windowed by multiplying them by the window function (Fig. 8c).

By applying our AO methods to the windowed fields, we obtained the aberrations at 5 × 5 positions (Fig. 8d). The relative aberrations to the center patch (Fig. 8e) clearly show how the aberrations vary spatially. We then reconstructed tomograms from the corrected fields for each aberration. The tomogram corrected with a single PSF could reveal the structure of the tissue within the patch (Fig. 8g) that had been hidden under the aberration (Fig. 8f). After collecting the tomograms corrected within each patch, the structure of the tissue across the entire FOV was revealed (Fig. 8h).

**Aberration correction for deep tissue imaging**

Aberration correction within deep tissue is a primary focus of AO. We analyze the performance of different AO methods for correcting aberrations at a specific depth within a sample using time-gated reflection imaging. For quantitative comparison, we employed a multi-layer light propagation simulation based on the optical parameters of biological tissue[51,52] (see Methods).

To evaluate AO methods, we imaged the target within the tissue model and achieved a tight focus at the target depth through wave-front shaping. Quantitative analysis was conducted by comparing

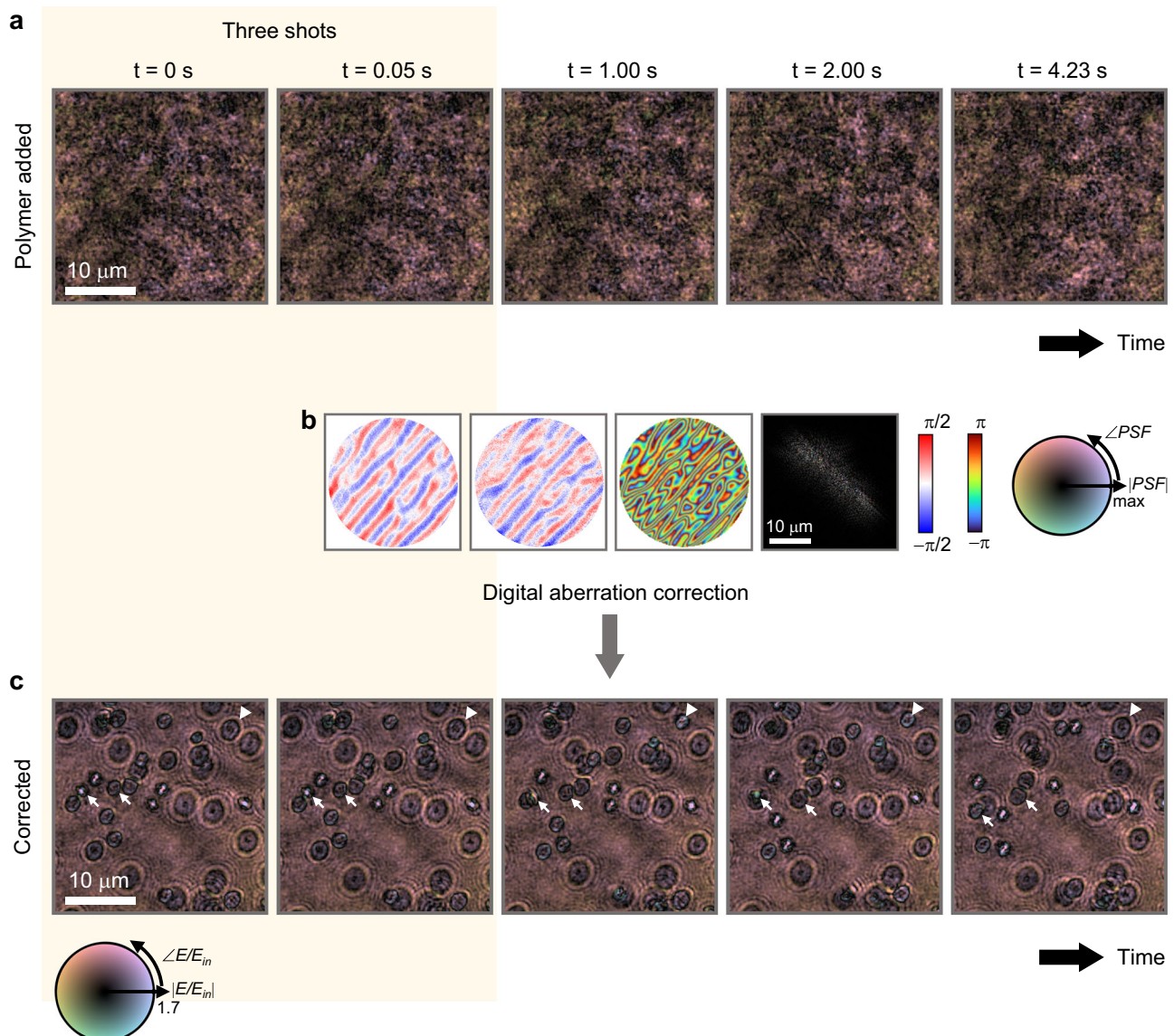

**Fig. 7 | Demonstration of aberration correction on moving 2-μm-diameter beads.**
**a** Sequential snapshots from a video captured at 60 fps, showing fields scattered by moving beads through the aberrating polymer. **b** Our correction method depicts the x- and y-components of the phase gradient, the phase of outgoing aberration function, and the corresponding PSF, demonstrating the separation of the aberration function from the scattered field can be conducted within three frames (indicated by yellow highlighting). **c** Corrected field sequences. The arrows track the lateral movement of two beads, and the triangle points to the axial movement of a single bead.

Strehl ratios, defined as the peak intensity of the focus relative to an ideal focus achieved with a guide star. To address aberrations from refractive index mismatches at interfaces, such as the tissue-medium or skull-brain boundaries, we incorporated smooth phase functions into the first layer to simulate light deflection at the tissue boundary (Supplementary Fig. 1e).

Figure 9 shows that different AO methods enhanced image and focus quality up to a mild surface deflection, as indicated by increased Strehl ratios (Fig. 9f). However, with a strong surface deflection, the isoplanatic patch size decreased below that of the PSF, leading to unreliable convergence of guide-star-free AO techniques (Fig. 9b, right column). In contrast, our method enhanced the Strehl ratio by detecting the average aberration within the windowed area, a benefit that stems from the analyticity of the aberration matrix (Fig. 9c).

Our approach enables aberration correction even when the PSF is larger than the isoplanatic patch, using an iterative windowing process.

This process involves (i) detecting aberrations from windowed fields, (ii) correcting the original fields, and (iii) reapplying the window, and repeating steps (i)–(iii). Combined with the aberration matrix, this method led to a 20-fold improvement in the Strehl ratio under strong aberration conditions (Fig. 9d; see Supplementary Fig. 2 for a visualization), while applying iterative windowing to existing computational methods does not yield improvements in this context. Beyond simulations, our method is compatible with experimental reflection matrix imaging data[53,54] (see Supplementary Fig. 3).

## Discussion
In this study, we present a computational AO method that is versatile across a wide array of sample types, including volumetric, dynamic, and those subject to strong scattering and aberrations. Our method stands out by effectively utilizing the angular memory effect, enabling more robust aberration detection than existing techniques. While existing computational AO methods struggle with transmission

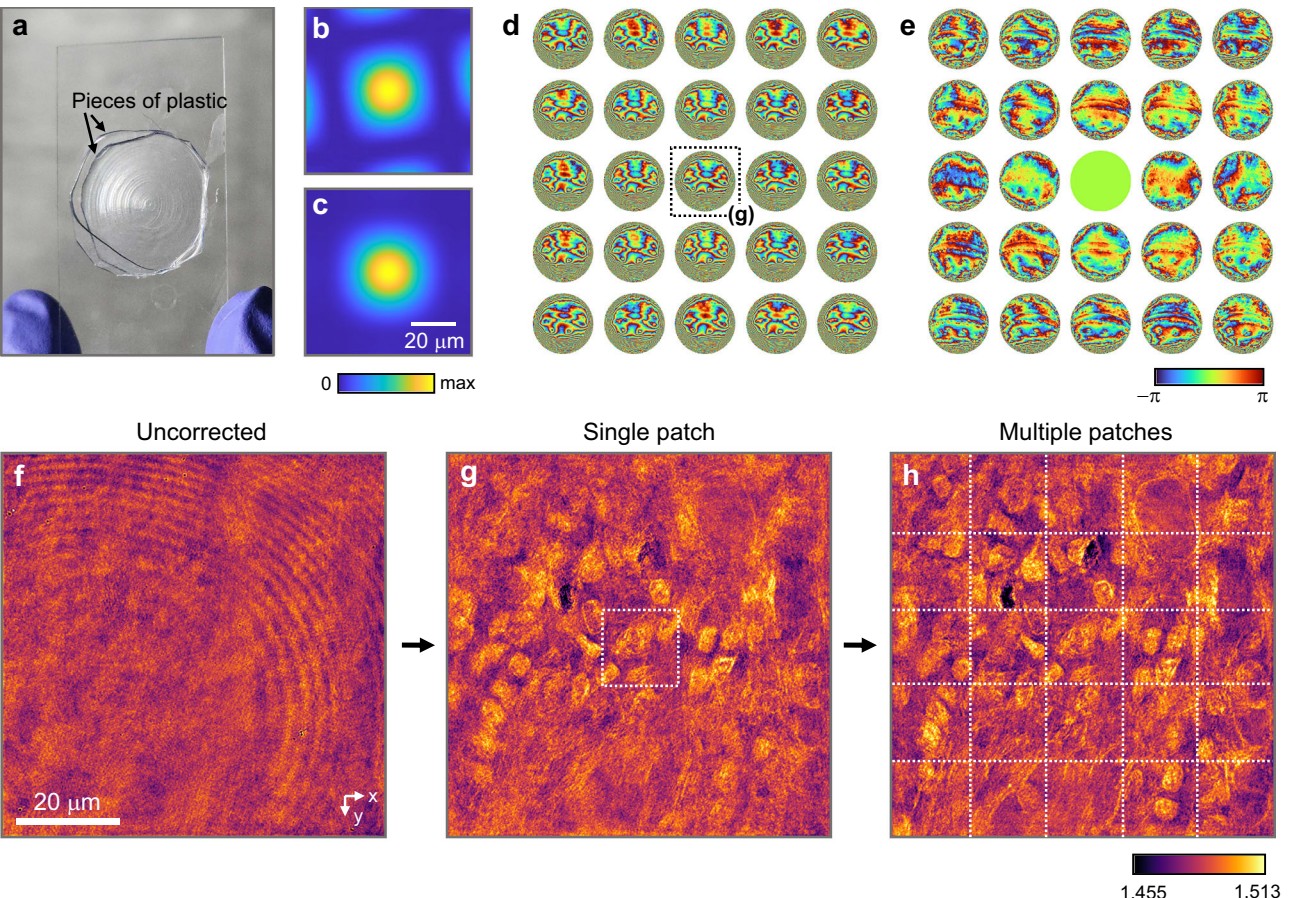

**Fig. 8 | Correction of spatially varying aberrations with 10-µm thick human tissue. a** 10-µm thick human pancreatic tissue sandwiched in between two pieces of plastic. **b** Window applied to the incoming wave basis. **c** Window applied to the outgoing wave basis. **d** Outgoing aberration functions determined using window functions centered on each patch. **e** Aberration functions subtracted from that of the central patch. **f** RI tomogram aberrated by plastic pieces. **g** RI tomogram corrected using the aberration function belonging to the central patch. **h** RI tomogram combined after correction for aberrations within each patch.

imaging of thick samples, we achieved aberration correction in tissues up to 100 µm thick by constructing the aberration matrix based on the high correlation within a limited angular range. The aberration matrix is also robust to temporal decorrelation of scattered light in dynamic samples, a common limitation in AO, using the relative phase of the consecutive images.

The limitations of our method are consistent with those of gradient-based wavefront sensing methods. Integrating the phase gradients converts white noise into Brownian noise, increasing the error in lower-order aberrations[55]. To improve accuracy, errors in the phase gradients should be sufficiently suppressed, requiring careful consideration of the number of incident angles and the magnitude of the tilt angle. In addition, we used an inverse gradient algorithm based on the least-square method, which neglects branch points (phase singularities) that generally appear in sample-induced aberrations in the multiple scattering regime. To address this discrepancy, the phase of the branch points needs to be obtained separately[56–58].

This work has primarily focused on applying the proposed method in transmission imaging, although AO shows promise in in vivo deep tissue imaging using a time-gated reflection-mode microscope. As a quick demonstration, we present simulated results of time-gated reflection imaging, showing the effectiveness of the aberration matrix under strong aberrations induced by upper tissue layers, compared to the existing computational methods. Moving forward, we aim to experimentally demonstrate our method for in vivo reflection imaging in time-gated reflection microscopy.

## Methods

### Human tissue samples

Tissue samples were obtained from Asan Medical Center with approval from the Institutional Review Board (IRB) with waiver of informed consent from patients [2021-1698] due to the retrospective nature of the study. The study adheres to the ethical principles delineated in the Declaration of Helsinki[59], emphasizing the utmost importance of protecting the rights and welfare of research subjects.

### Reweighting the aberration matrix to improve SNR

According to Eq. (5), the argument of the aberration matrix consists of the aberration functions, while its modulus indicates the angular spectrum of the scattered light:

$$\arg[A(\mathbf{k}_{out};\,\mathbf{k}_{in})] \approx \arg\left[e^{i\Delta\phi_{out}(\mathbf{k}_{out})}e^{i\Delta\phi_{in}(\mathbf{k}_{in})}\right] \tag{9}$$

$$|A(\mathbf{k}_{out};\,\mathbf{k}_{in})| \approx |T(\mathbf{k}_{out};\,\mathbf{k}_{in})|^2 \tag{10}$$

Since the SNR of the measured phase is proportional to its modulus, the modulus of the matrix can be used as a weight for effective factorization of the aberration matrix in the presence of noise. If the angular spectrum is sufficiently wide, as in reflection matrix imaging discussed in Supplementary Note 1, the aberration matrix can be directly factorized without changing the modulus.

In the case of transmission imaging, the high scattering anisotropy factor of biological tissues[60] results in a narrow angular spectrum

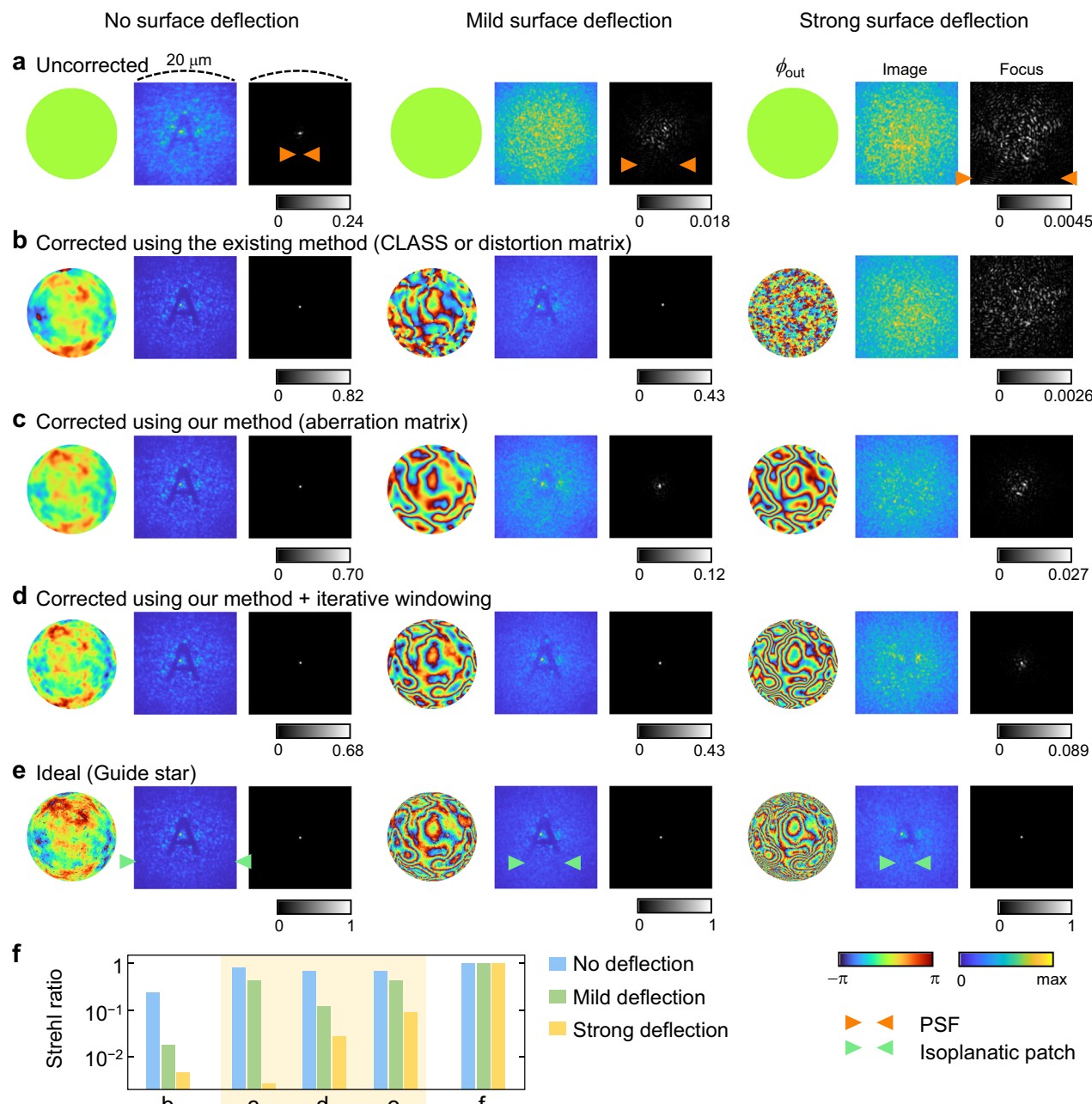

**Fig. 9 | Imaging and focusing through tissue model. a–e** Intensity images of the target object and foci at the target depth, corrected based on the outgoing aberration functions obtained using different methods. **f** Strehl ratios characterized by the peak intensity of the focus relative to that of the ideal focus. The sizes of the PSF and isoplanatic patch are indicated by orange and green triangles, respectively. Source data are provided as a Source Data file.

of transmitted light. Consequently, the diagonal elements are much more heavily weighted than the off-diagonal elements, leading to ineffective factorization due to the high matrix rank. A better strategy is to threshold the matrix elements above the noise floor, rather than simply utilizing the modulus as the weight:

$$A(\mathbf{k}_{out};\mathbf{k}_{in}) \leftarrow \begin{cases} e^{i\arg A(\mathbf{k}_{out};\mathbf{k}_{in})} & \text{if } |A(\mathbf{k}_{out};\mathbf{k}_{in})| > \text{threshold} \\ 0 & \text{otherwise} \end{cases} \quad (11)$$

We manually adjusted the threshold so that the argument of the matrix exhibits smooth phase gradients, thereby removing impulse noise. (see Supplementary Note 2).

In our experimental setup, the DMD notably acts as a significant source of the noise, producing multiple peaks in $k$-space[46] as shown in

Supplementary Note 5. Since these artifacts can exceed the threshold when the angular spectrum is narrow, we additionally mask the matrix elements far from the diagonal before factorization. For samples with lower scattering, such as in the 10- and 50-μm tissues, we limit the masking radii to 0.3–0.4 NA. In contrast, masking is not required for the 100-μm thick tissue sample.

**Factorization of aberration matrix**

In our method, incoming and outgoing aberration can be separated by factorizing the aberration matrix. The simplest approach is to perform SVD and retain only the first singular vectors. Although SVD provides aberration functions of considerable quality, we introduce an iterative method to further improve the accuracy. Using Eq. (9), which remains valid after reweighting, the following equations can be

derived:

$$\arg\left[(A^\dagger A)(\mathbf{k};\ \mathbf{k}')\right] \approx \arg\left[e^{-i\Delta\phi_{\text{in}}(\mathbf{k})}e^{i\Delta\phi_{\text{in}}(\mathbf{k}')}\right] \tag{12}$$

$$\arg\left[(AA^\dagger)(\mathbf{k};\ \mathbf{k}')\right] \approx \arg\left[e^{i\Delta\phi_{\text{out}}(\mathbf{k})}e^{-i\Delta\phi_{\text{out}}(\mathbf{k}')}\right] \tag{13}$$

where † denotes the conjugate transpose and matrix multiplications are used. To solve these equations, we convert them into the unimodular quadratic programming (UQP) problem:

$$\max_{\mathbf{v}} \mathbf{v}^\dagger \mathbf{M} \mathbf{v}$$
$$\text{subject to} |\mathbf{v}(\mathbf{k}')| = 1 \tag{14}$$

where $\mathbf{v}$ an optimization variable, $\mathbf{M}$ is $A^\dagger A$ or $AA^\dagger$, and the maximum point is at $\mathbf{v}(\mathbf{k}') = e^{-i\Delta\phi_{\text{in}}(\mathbf{k}')}$ or $\mathbf{v}(\mathbf{k}') = e^{i\Delta\phi_{\text{out}}(\mathbf{k}')}$, respectively. In this work, we solve the UQP using the projection power iteration, where $\mathbf{v}$ is iteratively updated starting from 1 as follows[61]: $\mathbf{v} \leftarrow e^{i\arg(\mathbf{Mv})}$.

Here, solving Eq. (13) is computationally more intensive than Eq. (12) because the aberration matrix typically has more rows than columns. Therefore, we solve Eq. (12) first and use the following formula, derived from Eq. (9), to obtain $\Delta\phi_{\text{out}}$:

$$\Delta\phi_{\text{out}}(\mathbf{k}_{\text{out}}) = \arg\left[\sum_{\mathbf{k}_{\text{in}}} A(\mathbf{k}_{\text{out}};\ \mathbf{k}_{\text{in}})e^{-i\Delta\phi_{\text{in}}(\mathbf{k}_{\text{in}})}\right] \tag{15}$$

## Implementation of matrix-based AO method

For comparison purposes, we have implemented the matrix-based guide-star-free AO method, CLASS[25] or distortion matrix[27], as follows. Given the plane wave incidence, Eq. (1) can be rewritten by taking the 2D Fourier transform:

$$\tilde{E}_{\text{out}}(\mathbf{k}_{\text{out}};\ \mathbf{k}_{\text{in}}) = \tilde{P}_{\text{out}}(\mathbf{k}_{\text{out}})\tilde{S}(\mathbf{k}_{\text{out}} - \mathbf{k}_{\text{in}})\tilde{P}_{\text{in}}(\mathbf{k}_{\text{in}}) \tag{16}$$

This equation leads to the following inequality for arbitrary functions $\phi_{\text{out}}$ and $\phi_{\text{in}}$:

$$\left|\sum_{\mathbf{k}_{\text{out}}-\mathbf{k}_{\text{in}}=\mathbf{K}} e^{-i\phi_{\text{out}}(\mathbf{k}_{\text{out}})}\tilde{E}_{\text{out}}(\mathbf{k}_{\text{out}};\ \mathbf{k}_{\text{in}})e^{-i\phi_{\text{in}}(\mathbf{k}_{\text{in}})}\right| \leq \sum_{\mathbf{k}_{\text{out}}-\mathbf{k}_{\text{in}}=\mathbf{K}}\left|\tilde{E}_{\text{out}}(\mathbf{k}_{\text{out}};\ \mathbf{k}_{\text{in}})\right|$$
$$= \sum_{\mathbf{k}_{\text{out}}-\mathbf{k}_{\text{in}}=\mathbf{K}}\left|\tilde{S}(\mathbf{k}_{\text{out}} - \mathbf{k}_{\text{in}})\right| \tag{17}$$

where equality holds if $\tilde{P}_{\text{out}}(\mathbf{k}_{\text{out}}) = e^{i\phi_{\text{out}}(\mathbf{k}_{\text{out}})}$ and $\tilde{P}_{\text{in}}(\mathbf{k}_{\text{in}}) = e^{i\phi_{\text{in}}(\mathbf{k}_{\text{in}})}$.

Thus, PSFs can be found by maximizing the following objective function:

$$L(\phi_{\text{out}},\ \phi_{\text{in}}) = \sum_{\mathbf{K}}\left|\sum_{\mathbf{k}_{\text{out}}-\mathbf{k}_{\text{in}}=\mathbf{K}} e^{-i\phi_{\text{out}}(\mathbf{k}_{\text{out}})}\tilde{E}_{\text{out}}(\mathbf{k}_{\text{out}};\ \mathbf{k}_{\text{in}})e^{-i\phi_{\text{in}}(\mathbf{k}_{\text{in}})}\right|^2 \tag{18}$$

It is noteworthy that maximizing this function is analogous to maximizing the intensity of the corresponding 2D image ($S(\mathbf{r})$) according to Parseval's theorem.

Since this maximization problem is a non-convex optimization problem, the final solution generally falls into different local maxima depending on the exact implementation.

Our implementation combines methods discussed in recent literature[32,33]. To obtain the PSFs, we iteratively maximize $L(\phi_{\text{out}}, \phi_{\text{in}})$

while fixing $\phi_{\text{out}}$ or $\phi_{\text{in}}$ alternatively:

$$\phi_{\text{in}} \leftarrow \arg\max_{\phi_{\text{in}}} L(\phi_{\text{out}}, \phi_{\text{in}}) = \arg\max_{\phi_{\text{in}}} \sum_{\mathbf{k}'_{\text{in}}, \mathbf{k}_{\text{in}}} e^{i\phi_{\text{in}}(\mathbf{k}'_{\text{in}})}$$
$$\left[\sum_{\mathbf{K}} D^*_{\text{in}}(\mathbf{K};\ \mathbf{k}'_{\text{in}})D_{\text{in}}(\mathbf{K};\ \mathbf{k}_{\text{in}})\right]e^{-i\phi_{\text{in}}(\mathbf{k}_{\text{in}})}, \tag{19}$$
$$D_{\text{in}}(\mathbf{K};\ \mathbf{k}_{\text{in}}) = e^{-i\phi_{\text{out}}(\mathbf{k}_{\text{in}}+\mathbf{K})}\tilde{E}_{\text{out}}(\mathbf{k}_{\text{in}}+\mathbf{K};\ \mathbf{k}_{\text{in}}),$$

$$\phi_{\text{out}} \leftarrow \arg\max_{\phi_{\text{out}}} L(\phi_{\text{out}}, \phi_{\text{in}}) = \arg\max_{\phi_{\text{out}}} \sum_{\mathbf{k}_{\text{out}}, \mathbf{k}'_{\text{out}}} e^{-i\phi_{\text{out}}(\mathbf{k}_{\text{out}})}$$
$$\left[\sum_{\mathbf{K}} D_{\text{out}}(\mathbf{k}_{\text{out}};\ \mathbf{K})D^*_{\text{out}}(\mathbf{k}'_{\text{out}};\ \mathbf{K})\right]e^{i\phi_{\text{out}}(\mathbf{k}'_{\text{out}})}, \tag{20}$$
$$D_{\text{out}}(\mathbf{k}_{\text{out}};\ \mathbf{K}) = e^{-i\phi_{\text{in}}(\mathbf{k}_{\text{out}}-\mathbf{K})}\tilde{E}_{\text{out}}(\mathbf{k}_{\text{out}};\ \mathbf{k}_{\text{out}} - \mathbf{K}),$$

where $D_{\text{in}}(\mathbf{K};\ \mathbf{k}_{\text{in}})$ and $D_{\text{out}}(\mathbf{k}_{\text{out}};\ \mathbf{K})$ are equivalent to the distortion matrix. The maximization problems in these equations are UQP and can be solved using the projection power iteration, similar to the previous section and the study of Bureau et al.[32]. When the iteration converges, the PSFs are determined from $\phi_{\text{out}}$ and $\phi_{\text{in}}$ as $\tilde{P}_{\text{out}}(\mathbf{k}_{\text{out}}) = e^{i\phi_{\text{out}}(\mathbf{k}_{\text{out}})}$ and $\tilde{P}_{\text{in}}(\mathbf{k}_{\text{in}}) = e^{i\phi_{\text{in}}(\mathbf{k}_{\text{in}})}$.

## Spatially varying aberrations and windowing

When a plane wave passes through an aberrating medium, its angular spectrum deviates from the incident angle due to scattering. Since the correlation function and the spectral density are related by the Fourier transform, the correlation length of the scattered wave decreases inversely proportional to the bandwidth of the angular spectrum:

$$l_c \approx \frac{1}{\sqrt{\langle(\sin\theta/\lambda)^2\rangle}} \tag{21}$$

where $\theta$ is the total scattering angle and $\lambda$ is the wavelength in the medium. The correlation can be further related to the scattering parameter of the medium:

$$l_c \approx \frac{\lambda}{\sqrt{2}}\sqrt{\frac{l_t}{L}} \tag{22}$$

where $l_t$ and $L$ are the transport mean free path and thickness of the medium, respectively, and the identity $\langle\cos\theta\rangle = e^{-L/l_t}$, and the small angle approximation, $\sin^2\theta \approx 2 - 2\cos\theta$, are used for the derivation. This translational correlation is the origin of isoplanatic patch, which is considerably larger than the wavelength in the anisotropic scattering regime ($L \ll l_t$)[34,35,41].

To utilize this correlation, AO models typically incorporate window functions applied to incoming and outgoing waves. By properly windowing the waves, the effect of each aberrating medium can be described by a single PSF near the windowed region.

In our demonstration, we divided the entire FOV into patches and applied Gaussian windows with $\sigma = 11$ mm, centered on each patch, to obtain PSFs within each patch. First, outgoing waves were multiplied by window functions $W_{\text{out}}(\mathbf{r}_{\text{out}})$ at each patch. In contrast, numerically applying window functions to incoming waves was not a trivial task due to the sparsely sampled incident angles. To mitigate this issue, we focused on the Fourier representation of the windowed plane wave:

$$W_{\text{in}}(\mathbf{r}_{\text{in}})e^{i\mathbf{k}_{\text{in}}\cdot\mathbf{r}_{\text{in}}} = \sum_{\mathbf{k}} \tilde{W}_{\text{in}}(\mathbf{k})e^{i(\mathbf{k}_{\text{in}}+\mathbf{k})\cdot\mathbf{r}_{\text{in}}} \tag{23}$$

where $\tilde{W}_{\text{in}}(\mathbf{k})$ is the Fourier coefficient of $W_{\text{in}}(\mathbf{r}_{\text{in}})$.

This equation indicates that windowing can alternatively be implemented by additionally sampling the incident vectors near $\mathbf{k}_{\text{in}}$ and superposing the corresponding outgoing fields. Thus, the overall

windowing process can be performed by replacing the measured fields with the following windowed fields,

$$E_{\text{windowed}}(\mathbf{r}_{\text{out}}; \mathbf{k}_{\text{in}}) = W_{\text{out}}(\mathbf{r}_{\text{out}}) \sum_{\mathbf{k}} \tilde{W}_{\text{in}}(\mathbf{k}) E(\mathbf{r}_{\text{out}}; \mathbf{k}_{\text{in}} + \mathbf{k}) \quad (24)$$

where AO models can now be applied.

In the experiment, we additionally scanned the incident vectors over a $3 \times 3$ rectangular grid with a scanning interval $\Delta k_{\text{scan}} = 2\Delta k$ near each point shown in Fig. 3c. To center the window function at the specific point $\mathbf{r}_{\text{c}}$, we determined the phase of $\tilde{W}_{\text{in}}(\mathbf{k})$ by matching the phase of the windowed waves at the desired region:

$$\arg[\tilde{W}_{\text{in}}(\mathbf{k}) E(\mathbf{r}_{\text{c}}; \mathbf{k}_{\text{in}} + \mathbf{k})] = \arg[\tilde{W}_{\text{in}}(\mathbf{0}) E(\mathbf{r}_{\text{c}}; \mathbf{k}_{\text{in}})] \quad (25)$$

In addition, we determined the modulus of $\tilde{W}_{\text{in}}(\mathbf{k})$ for a Gaussian-like shape as follows:

$$\begin{aligned} |\tilde{W}_{\text{out}}(|\mathbf{k}| = 0)| &= 4, \\ |\tilde{W}_{\text{out}}(|\mathbf{k}| = \Delta k_{\text{scan}})| &= 2, \\ |\tilde{W}_{\text{out}}(|\mathbf{k}| = \sqrt{2}\Delta k_{\text{scan}})| &= 1 \end{aligned} \quad (26)$$

We applied our AO method to the windowed fields for each patch, and then corrected the measured field using the detected aberrations. The corrected fields were utilized to reconstruct the tomograms, which were then cropped within each patch and stitched together. When stitching the tomograms, RI offsets, resulting from the global phase ambiguity of the PSFs, must be compensated between nearby tomograms. In this context, a slowly varying background can appear in the stitched tomogram if the windows are not much smaller than the patch. To remove this background, we filtered the low-frequency component with a Gaussian filter of $\sigma = 1/4\ \mu\text{m}^{-1}$.

## Tomographic reconstruction using gradient-based Rytov approximation

Rytov approximation is one of the approximations in the elastic scattering theory, which is valid when the gradient of the phase is sufficiently smaller than the scattering potential. Because it can be applied to thick samples, the Rytov approximation is often favored over the Born approximation for tomographic reconstruction of biological tissues.

The Rytov approximation describes the scattering of a scalar wave under plane wave incidence using the following formula[62]:

$$E(\mathbf{r}_{3D})/E_{\text{in}}(\mathbf{r}_{3D}) = \exp\left[(G(\mathbf{r}_{3D})/E_{\text{in}}(\mathbf{r}_{3D})) * V(\mathbf{r}_{3D})\right] \quad (27)$$

where $\mathbf{r}_{3D} = (\mathbf{r}, z)$ represents the three-dimensional coordinates of the position space, $E(\mathbf{r}_{3D})$ is an outgoing wave, $E_{\text{in}}(\mathbf{r}_{3D})$ is an incoming plane wave with unit modulus, $G(\mathbf{r}_{3D})$ is the Green's function for Helmholtz equation, and $V(\mathbf{r}_{3D})$ is the scattering potential of the sample.

In tomographic reconstruction, Eq. (27) is inverted to derive the scattering potential from the outgoing waves. However, this is not a trivial problem, as the exponential function in a complex domain is a many-to-one function. This can be manifested by taking the complex logarithms on both sides of Eq. (27):

$$\log|E(\mathbf{r}_{3D})/E_{\text{in}}(\mathbf{r}_{3D})| + i \arg(E(\mathbf{r}_{3D})/E_{\text{in}}(\mathbf{r}_{3D})) = (G(\mathbf{r}_{3D})/E_{\text{in}}(\mathbf{r}_{3D})) * V(\mathbf{r}_{3D}) \quad (28)$$

As the argument is a multivalued function, it adds ambiguity when reconstructing tomograms. The conventional way to alleviate this ambiguity is to apply the phase unwrapping to the argument of

$E(\mathbf{r}_{3D})/E_{\text{in}}(\mathbf{r}_{3D})$, which is justified by the assumption of its small gradient.

However, the unwrapped phase is not generally unique when phase residues are present[63], which is a common occurrence when imaging thick samples. These additional phase residues can also emerge after aberration correction, particularly at locations where the intensity is low. Consequently, phase unwrapping alone cannot provide a well-defined formula for tomographic reconstruction.

Instead of phase unwrapping, the small gradient assumption can be directly utilized by taking the gradient of Eq. (28) and choosing the smallest value. Consequently, we now obtain the well-defined inversion formula of Rytov approximation as follows:

$$\nabla \log|E(\mathbf{r}_{3D})/E_{\text{in}}(\mathbf{r}_{3D})| + i\,\text{wrap}[\nabla \arg(E(\mathbf{r}_{3D})/E_{\text{in}}(\mathbf{r}_{3D}))] = (G(\mathbf{r}_{3D})/E_{\text{in}}(\mathbf{r}_{3D})) * \nabla V(\mathbf{r}_{3D}) \quad (29)$$

where wrap$[\cdot]$ is a wrapping function that wraps input values into the interval $(-\pi, \pi]$.

By deconvolving the Green's function in 3D, the scattering potential, and the corresponding RI tomogram can be reconstructed from the outgoing waves.

## 3D phase correlation for evaluating 3D PSFs

We have introduced a 3D phase correlation to quantitatively evaluate the PSFs before and after aberration correction. Here, we explain the formula and meaning of the 3D phase correlation in detail. The 3D phase correlation of 3D fields $E_1(\mathbf{r}_{3D})$ and $E_2(\mathbf{r}_{3D})$ is defined as follows:

$$\left| \frac{\text{IFT3D}\{\exp\left[i\arg(\tilde{E}_1(\mathbf{k}_{3D})\tilde{E}_2^{*}(\mathbf{k}_{3D}))\right]\delta(|\mathbf{k}_{3D}| - k_{\text{medium}})\}(\mathbf{r}_{3D})}{\text{IFT3D}[\delta(|\mathbf{k}_{3D}| - k_{\text{medium}})](\mathbf{r}_{3D} = \mathbf{0})} \right|^2 \quad (30)$$

where $E(\mathbf{r}_{3D})$ is numerically propagated from the 2D field $E(\mathbf{r})$ using the angular spectrum method, $\tilde{E}(\mathbf{k}_{3D})$ represents the 3D Fourier transform of $E(\mathbf{r}_{3D})$, IFT3D denotes the 3D inverse Fourier transform, and $k_{\text{medium}}$ is the magnitude of the wave vector of the given medium and wavelength.

When both fields have been scattered from the same sample, the correlation of the fields reduces to the correlation of their corresponding PSFs since

$$\exp\left[i\arg(\tilde{E}_1(\mathbf{k}_{3D})\tilde{E}_2^{*}(\mathbf{k}_{3D}))\right] = \tilde{P}_1(\mathbf{k}_{3D})\tilde{P}_2^{*}(\mathbf{k}_{3D}) \quad (31)$$

where $\tilde{P}_1(\mathbf{k}_{3D})$ and $\tilde{P}_2(\mathbf{k}_{3D})$ are the 3D Fourier transforms of the PSFs of fields $E_1(\mathbf{r}_{3D})$ and $E_2(\mathbf{r}_{3D})$, respectively, assuming their moduli are unity.

In the main text, we specifically set $E_1(\mathbf{r}_{3D})$ to be the aberrated or corrected fields, and $E_2(\mathbf{r}_{3D})$ to be the polymer-free fields. By neglecting system-induced aberrations in the polymer-free fields, $\tilde{P}_2(\mathbf{k}_{3D})$ becomes 1, simplifying the 3D correlation to the 3D intensity PSF of the aberrated or corrected fields, $|P_1(\mathbf{r}_{3D})|^2$, normalized by the ideal PSF.

## Multi-layer light propagation simulation for performance quantification of different AO methods

To validate AO methods in deep tissue imaging, we simulated 3D light transport in biological tissue. To reduce the computational complexity, we divided the tissue volume into parallel layers equally spaced at $10\ \mu\text{m}$ intervals. The transmission and reflection coefficients of each layer, shown in Supplementary Fig. 1a, were determined such that the layers exhibit typical scattering parameters of biological tissues, with the scattering and transport mean free paths are set to $l_{\text{s}} = 53\ \mu\text{m}$ and $l_{\text{t}} = 1.1\ \text{mm}$, respectively[64].

We calculated the field of light propagating in the forward ($+z$) direction using the beam propagation method (BPM), which is widely used for simulating anisotropic scattering media[51,52]. Specifically, the field incident on one layer is multiplied by the transmission coefficient of the layer and then propagated to the next layer in turn. To obtain the

field of reflected light, we multiplied the forward propagating fields by the reflection coefficients at each layer, and then used BPM in the opposite direction to account for backward propagation. The effect of multiple reflections between layers can be ignored as backward scattering is ignored in BPM due to the low probability of backward scattering in anisotropic media (reflectance <0.04 in the simulation).

To compare the performance of the proposed and existing computational AO methods, we used the same number of incident waves as shown in Supplementary Fig. 1b. The window functions shown in Supplementary Fig. 1c were used to select the target patch in which the letter 'A' is embedded. The incident waves were pre-windowed to reduce computational cost. To implement the time-gating technique, the reflected fields calculated at 40 wavelengths were summed with the spectral weights shown in Supplementary Fig. 1d, resulting in an axial resolution of 11 μm. The focal plane and reference mirror were positioned at the target depth, $z = 2l_s = 100\,\mu\text{m}$.

To achieve focusing through the tissue, we applied wavefront correction to the pupil plane so that the pupil function counteracts the incoming aberration function before incidence on the sample. Given that the k-space of incoming waves is sparsely probed, we estimated the incoming aberration function from the outgoing aberration function using time-reversal symmetry, $\phi_{\text{in}}(\mathbf{k}) = \phi_{\text{out}}(-\mathbf{k})$, rather than extracting it directly from the aberration matrix.

## Reporting summary

Further information on research design is available in the Nature Portfolio Reporting Summary linked to this article.

## Data availability

Data of 10-, 50-, and 100-μm thick human tissues presented in the main text and the configuration for the simulations in Supplementary Information are available from our GitHub repository: https://github.com/BMOLKAIST/AberrationMatrix[65]. All other data is available from the corresponding author upon request due to its large size. Requests will be fulfilled within 5 weeks. Source data are provided with this paper.

## Code availability

The supporting codes for this article, including codes for computational AO based on the proposed and existing methods, tomographic reconstruction, and simulation of time-gated reflection imaging, are available from GitHub repository: https://github.com/BMOLKAIST/AberrationMatrix[65].

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

## Acknowledgements

This work was supported by National Research Foundation of Korea grant funded by the Korea government (MSIT) (RS-2024-00442348, 2015R1A3A2066550, RS-2022-NR068141), Institute of Information & communications Technology Planning & Evaluation (IITP; 2021-0-00272) grant funded by the Korea government, the Ministry of Trade, Industry and Energy (MOTIE) and Korea Institute for Advancement of Technology (KIAT) through the International Cooperative R&D program (P0028463), and the Korean Fund for Regenerative Medicine (KFRM) grant funded by the Korea government (the Ministry of Science and ICT and the Ministry of Health & Welfare) (21A0101L1–12).

## Author contributions

C.O. developed the theory and analyzed the data. C.O., H.H., and M.L. conducted the experiments. Y.P. supervised the project. All authors wrote the manuscript.

## Competing interests

The authors declare no competing interests.
