## [Transparent Peer Review file · Nature Communications]

Digital aberration correction for enhanced thick tissue imaging exploiting aberration matrix and tilt-tilt correlation from the optical memory effect

Corresponding Author: Professor YongKeun Park

Version 0:

Reviewer comments:

Reviewer #1

(Remarks to the Author)

This manuscript introduces a novel digital aberration correction technique based on the angular optical memory effect. By computing the phase differences between the light field of different incident angles, the authors were able to extract the phase information to correct for the aberration from the optical system and an artificially introduced aberrating optical element. Using a transmission-mode holotomography setup, the authors validated their approach experimentally. There are multiple areas where substantial improvements are necessary to meet the publication standards of Nature Communications. The review below outlines specific aspects that, if addressed, could potentially enhance the clarity, rigor, and contribution of this work to the optics community.

Major Concerns:

The work in this manuscript motivated by the need to image structure within optically dense samples, such as biological tissues. However, the primary demonstrations use an “aberrating layer”, consisting of two lenses and a phase object, to introduce a “controlled aberration”. The nature and significance of the aberrations introduced by this layer should be quantified, as the relevance and effectiveness of the author’s technique to the spatially continuous aberrations found in biological samples is unclear. Comparative experiments, like those in the cited literature such as Kang et al 2015 (Nature Photonics), Kim et al 2019 (Nature Communications), and Jo et al 2022 (Science Advance), would substantially strengthen the paper. These prior works demonstrated aberration correction in biological tissue, including imaging beads embedded in thick rat tissue and in vivo imaging, and provides a much clearer demonstration of aberration correction.

Other questions and comments:

1. Mathematical model for aberration and scattering.

a. “To account for the volumetric properties of a sample, we generalize Eq. (3) by introducing a linear operator T to represent the scattering from the sample, rather than: $S(r)$ ”.

In Eqn (4), is r defined on 2- or 3-dimensional space?

b. Using the expansion in Eqn 5, the resulting Eqn 8 is independent of the linear operator T . It seems that ultimately the approach described in this paper is reduced to 2-dimensional. According to “Finally, the aberration is digitally corrected through the deconvolution of the outgoing fields E_{out} with the retrieved PSF P_{out} ”, it seems that the author’s approach only corrected P_{out} . If this is indeed the case, the authors should provide experimental evidence showing the advantages of the proposed approach over the traditional method that accounts for both P_{out} and $S(r)$, as stated in the main text (Eqn 3).

c. Clarity: The detail of the algorithm is split into three parts, some in the main text, some in the method, and some in the supplementary material, making it hard to follow and understand the overall approach. Improving the organization and presentation is needed to make the manuscript more accessible to readers.

2. Aberration correction with a 10- μ m thick human tissue.

a. “Intriguingly, the aberration’s phase accurately reflected the polymer’s wavy texture, as shown in Fig. 2b.”

Please explain how the phase accurately reflects the polymer’s texture.

b. According to the text “after polymer removal, un-aberrated fields still exhibit system-induced aberrations...”. Thus the

“without polymer” images were not corrected for “system-induced aberration”. Nominally, we expect the aberration-corrected image to be much sharper than the images taken “without polymer”, as the correction should apply to both system aberration and sample-induced aberration. Yet, from the image and according to the author’s interpretation, the aberration correction mainly corrects the aberration introduced by the polymer, which is external to the sample per se.

c. Does applying the system aberration correction obtained in Supplemental Figure 3 to the “without polymer” images here improve the image sharpness?

d. It would be easier to compare the detailed differences between the “corrected” images and the “without polymer” images if the overlay is shown.

e. Figures 3 and 4: More quantitative metrics should be used to quantify the aberration correction. The Fourier spectrum shown as the insert in Figure 5a should also be included. What is the axial resolution? How does it get improved after aberration correction? Can the 3D PSF be estimated?

f. Figure 3 legend: What does $\Delta u = 1.03 \lambda^{-1}$ mean? Why does it appear to be much larger than the value mentioned in the main text about Figure 5 “The memory effect range, defined by the 1/e correlation threshold, is quantified as a shift $0.024 \lambda^{-1}$ in the incident spatial frequency”? Presumably, Δu is related to the thickness of the sample, so why do Figure 3 and Figure 4 share the same Δu value?

3. Aberration correction with a 50- μm thick human tissue.

a. “This variation is attributed to using different polymer regions for each set of experiments (Figs. 3-5), as the polymer was manually positioned and removed for each trial, and the misfocusing of the sample.” What is the “misfocusing of the sample”? Can it be fixed by moving the objectives toward / away from the sample? It seems that the authors did not attribute the differences in the phase gradient to the increased thickness of the sample.

b. “This variation is attributed to using different polymer regions for each set of experiments (Figs. 3-5), as the polymer was manually positioned and removed for each trial, and the misfocusing of the sample.” Why does the polymer need to be removed after each trial? If the polymer does not need to be removed, do the author expect a largely similar phase gradient when switching from a 10 μm sample to a 50 μm sample?

c. Figure 4f: It would be clearer if the author could outline the position of the vessels and fibers.

4. Memory effect in a 100 μm thick human tissue and its influence on aberration correction

a. Figure 5b also suggests that the proposed technique corrects aberration induced by the polymer but not in the biological sample.

5. Aberration correction with moving samples:

a. “Our aberration correction method is adept at handling samples in motion, as long as their movement is slower than one-third of the camera’s frame rate”. The unit of movement should be [Length / Time] instead of [1/Time]. Additional discussion/analysis on the speed of the sample movement should be added.

b. “By measuring the components of the gradient separately, we can further extend our method’s applicability to dynamics approaching half the camera’s frame rate.” How to “measure the components of the gradient separately”? How does it differ from using the second measurement as the reference instead of the first one?

c. “After correcting for aberration, the fields accurately represent the beads’ shape, facilitating the analysis of bead dynamics at frequencies up to 20 Hz”: Why not at the full 60 Hz frame rate? Is the system actively correcting for the aberration using every 3 measurements?

6. Discussion and conclusion

a. “The strategic utilization of slight tilts in incidence angles within the angular memory effect’s range allows for more robust aberration measurement than existing techniques”. The author should provide quantitative evidence/comparison experiments to justify this claim.

7. Supplemental material section 2:

a. “In Methods section, we introduced a technique for directly integrating the phase gradients of the aberration. This method might necessitate phase unwrapping, which can pose challenges, particularly when working with masked phases. To address this issue, we can utilize the complex exponential of the phase instead of the phase itself.” The shared in “demo_code.m” implemented the method described in the supplemental material. It is not clear whether all the figures and experimental results shown in the main text were based on the algorithm in the supplemental material or in the main text (and the Method section). If they are based on the algorithm in the supplemental material, it should be clarified and integrated into the main text. Otherwise, the author should evaluate the difference in aberration correction from both algorithms.

8. Supplemental Figure 3: This figure and section are about correcting system-induced aberrations. So

a. Why a 10 μm thick human sample is used for correcting “system-induced aberration”? Why not use fluorescence beads for such system aberration correction?

b. Are the authors assuming that the aberration introduced by the 10 μm thick human sample is negligible? If yes, then why do the resulting “phase of the aberration in k-space” and the “corresponding PSF” shown in Supplementary Figure SF3a,b look so different? Especially for SF3a, there appear to be complex detailed structures in the PSF. Is this expected? Are FOV 1 and 2 just taken at different positions of the sample?

9. The authors indicated that the main text includes “a complete, detailed description of the code’s functionality (i.e. pseudocode)” in the “Code and Software Submission Checklist”. However, it seems that neither the pseudocode nor a summarizing description of the entire algorithm could be found in the main text.

10. The authors also provided MATLAB implementation and test data, which helps the reader to better understand their approach. However, the author only provided part of the code. As a manuscript focused on novel methodology, providing the dataset and the code for replicating the figures is essential. Regarding the "demo_code.m" in the provided Google Drive link:
- The tilted angles are [-0.8857, 0.1133] and [0.1167, 0.8590]. How were these tilted angles determined? They are not exactly orthogonal (dot product -0.006). Is this a numerical error? Alignment error?
 - The ``mk_ellipse`` function is missing, without which the demo code cannot be run.
 - What is ``noise`` and how is its value 200000 determined? If other laboratories want to replicate the measurement and technique, how should they determine this value?
 - What is ``mask_size`` and how was its value 0.2 determined? How does it affect the aberration correction result?

Conclusion:

This manuscript addresses a topic of interest to the field of optics. Yet from the image and according to the author's interpretation, it seems that aberration correction mainly corrects the aberration introduced by the polymer, which is external to the sample per se. This is a fundamental weakness of the current manuscript.

Reviewer #2

(Remarks to the Author)

This paper deals with the compensation of aberrations in a transmission-mode holotomography setup. Exploiting a tilt-tilt memory effect, the authors are able to compensate for aberrations induced by a polymer layer in a plane conjugated with the pupil plane of the microscope objective in transmission. More precisely, the method consists in extracting the phase derivative of the aberration phase law by comparing the wave-fields transmitted by the sample and aberrating layer when slightly tilting the direction of the incident plane wave. The choice of this tilt is subject to the following compromise: large enough such that noise does not hamper the numerical integration of the aberration phase law; small enough to benefit from the memory effect that scales as λ/L , with L the thickness of the sample. Based on this method, the authors manage to unscramble the aberrations induced by the polymer layer and retrieve a complex field close to the reference wave-field measured without the coating layer. The compensation of aberrations is also illustrated by building refractive index tomograms using gradient based Rytov approximation.

Although the method is technically interesting and appealing, its range of application seems restricted to the compensation of optical system aberrations. Yet, the real problem for deep imaging in tissues is not the aberrations of the imaging system for which a wealth of correction methods already exists but the correction of sample-induced aberrations. Such aberrations cannot be compensated by the method proposed by the authors, or at least they do not prove it and even not discuss about this possibility. In that view, the abstract and introduction of the paper are misleading since the authors claim that their method "enhances imaging of thick human tissues" (quoted from the abstract). A correct statement would be that the method allows the compensation of far-field aberrations induced by a thin phase screen. For these reasons, I cannot support publication in Nature Communications for the paper in its current form. However, as I said, the proposed method is interesting and could deserve publication in more specific journal.

Please find below more detailed comments:

1/ One major flaw of the paper is the confusion about the "thick tissue" application that can let the inexperienced reader think that this method is dedicated to the compensation of sample-induced aberrations and multiple scattering. I suggest the authors to be more modest about the range of application of this work. The real challenge for deep imaging in transmission is solving the inverse problem that links the transmitted wave-field with the refractive index distribution as mentioned briefly by the authors in the conclusion (page 15, line 3) : "though challenges in tomogram reconstruction at this thickness highlight areas for future improvement."

2/ Just before, the authors claim: "This approach surpasses traditional methods in handling thick tissue samples". This assertion shall be supported by a quantitative comparison with traditional methods. For instance, a comparison with CLASS algorithm that is based on the same idea (tilt-tilt memory effect) would be relevant. As they measured the transmission matrix across the sample, the correlation methods that the authors mention in the introduction could be also implemented in post-processing for comparison purpose.

3/ To be more quantitative, could the authors use a metric that will allow to compare the image they obtain with the ground truth image obtained without the polymer layer? For unexperienced eyes, it is difficult to assess the image quality in the different cases shown by the authors.

4/ The authors fail in showing in which practical situations this method can be useful. For instance, in the present case, the method addresses the aberrations induced by the polymer layer included on purpose in a very specific plane of the imaging system (output pupil plane of the MO). However, it does not seem to compensate for spherical aberrations induced by the imaging system itself. For instance, for the results shown in Fig.4, the authors had to refocus their tomogram by +3.5 μm to compensate for defocus aberration. Hence it means, that their approach is not able to compensate for this basic aberration that seems to be quite common in the field. Hence, the question is: For which kind of aberrations the method is good for?

5/ Related to that questions: If the aberrating layer is placed between the microscope objective and the sample, can we expect the method to work to some extent? Or is the method restricted to pupil aberrations?

6/ The authors said that their method is robust with respect to motion of the sample. I guess this is not the case for time-varying aberrations, it may be fair to specify it.

7/ One interesting feature of the proposed method is the ability to discriminate between input and output aberrations. However, the paper does not demonstrate this feature since the polymer layer is only included in the output arm.

Besides these questions about the scope of this work, please find below some more specific points:

8/ Page 5, line 16: "It is worth noting that the linear operator T is equivalent to the scattering matrix of a sample, which comprises the transmission matrix and reflection matrix^{38,39}."

The linear operator is equivalent to the transmission matrix, not to the whole scattering matrix.

9/ I have some doubt about the way the memory effect is included in the analytical model presented by the authors. There is no impact on the method but it may be misleading for readers. In Eq. 5, the angular memory effect is described as follows:

$$T_{\mathbf{k}_{in}+\Delta\mathbf{k}}(\mathbf{k}+\Delta\mathbf{k})=T_{\mathbf{k}_{in}}(\mathbf{k})+O(\Delta\mathbf{k}/k_{MER})$$

This expression suggests that the transmitted wave-field increases with $\Delta\mathbf{k}$, which is non-sense. A normalization term is needed to avoid such inconsistency. According to me, a proper way to include the memory effect would be to introduce the correlation function, $C(\Delta\mathbf{k}/k_{MER})$, characteristic of the memory effect between $T_{\mathbf{k}_{in}+\Delta\mathbf{k}}(\mathbf{k}+\Delta\mathbf{k})$ and $T_{\mathbf{k}_{in}}(\mathbf{k})$, such that

$$C(\Delta\mathbf{k}/k_{MER})=\langle T_{\mathbf{k}_{in}+\Delta\mathbf{k}}(\mathbf{k}+\Delta\mathbf{k}) T_{\mathbf{k}_{in}}^*(\mathbf{k}) \rangle$$

The bias made on the estimation of the aberration law could be derived by considering this correlation function, which is actually plotted in Fig.5.

10/ There is a problem with equation numbers in the Methods section. The authors are referring to an equation number that does not correspond to the equation they want to refer to. For instance, page 27, line 3, they refer to Eq.14 to demonstrate Eq.14. The same problem appears further. Please double check all the equation references.

Reviewer #3

(Remarks to the Author)
Dear Editor and authors,

Correcting optical aberrations in a thick sample is challenging. Typically, it requires to insert a "guide star" within the sample or using an iterative wavefront correction. Those solutions are not appropriate for all samples due to the temporal decorrelation or due to the impossibility to insert non-invasively a guide star. In this article, the authors proposed and demonstrate a new approach to measure aberrated wavefronts that doesn't require a guide star or a long iterative feedback correction in a holographic microscope configuration. Thanks to the angular memory effect of the sample, illuminating a sample by a couple of planes waves separated by a small angle leads to the measurement of the derivative of the aberrated wavefront. The wavefront obtained then is used to digitally correct holographic images. The authors applied their approach in two proof-of-concept scenarios: for 3D refractive index imaging and holographic imaging of dynamic samples, where a strongly aberrated layer is added after the sample.

The concept is elegant, and the proof-of-concept experiments are convincing. But the applicability seems limited because some theoretical derivations aren't clear and because the limitations of the techniques aren't much discussed. Some additional points need to be clarified.

At first, I'm a bit confused with the notations used by the authors. In equation (4), T is defined as an equivalent to the scattering matrix of the sample. But if T is a complex matrix, what allow the authors to go from a matrix product to an Hadamard product? I feel an assumption is missing here that will help the reader to understand the limitation of this approach.

I feel there is a $\Delta\mathbf{k}$ range where the wavefronts gradients can be successfully reconstructed (whatever the angular memory effect of the sample). If $\Delta\mathbf{k}$ is too small, the sensitivity might be too low to measure the gradient in presence of noise. If it's too large the derivative might be misestimate. Can the authors precise/quantify the $\Delta\mathbf{k}$ range validity of their approach? To go further, this approach requires some angular memory effect. How much? The authors touched this issue with the 100um thick sample in figure 5. But it would be interesting to understand better. Can the authors link the memory effect range needed to the $\Delta\mathbf{k}$ range validity of their approach to estimate the tissue thickness they will be able to image?

To distinguish between incident and outgoing aberrations, the authors assumed that the incident aberrations are the same for different incident wave vectors. It's equivalent to assume that the incident aberrations are only due to the optical system. Is it always true? In a realistic scenario, where the aberrations to correct are induced by the sample itself, both incident and outgoing aberrations will depend on the incident wave vectors. Will the approach suggested by the authors still work?

Some phrasings are misleading. I quote from the Discussion and Conclusion: "It achieves effective aberration correction in samples up to 100 μm thick", the authors didn't show these experiments. They only show an aberration measurement through a 100 μm tissue with an indirect validation.

Additionally, the optical configuration used to image the dynamic samples isn't specified (figure 6). I believe it wasn't an holotomography measurement as the dynamic of the sample should have prevented to obtain the 76 images for 3D reconstruction in a good condition.

Minor comments:

- To justify, I quote "Furthermore, the corrected tomogram showcased enhanced axial resolution of red blood cells, as indicated by white arrows in the xz section, surpassing that of the tomogram derived from unaberrated fields.", the authors might want to add cross section of the tomograms to illustrate.
- In figure 5, how is define the field correlation? Is it the norm of $C(\Delta k)$?
- In SI "Tomographic reconstruction using gradient based Rytov approximation", the equation number on the text is off by 1. For example, in the sentence "In tomographic reconstruction, Eq. (17) is inverted to derive the scattering potential from the ..." I believe we should read Eq. (16) instead.

Reviewer #4

(Remarks to the Author)

Version 1:

Reviewer comments:

Reviewer #1

(Remarks to the Author)

The authors have significantly improved the clarity and organization of the manuscript. This work represents an advance in computational adaptive optics. However, its practical value for imaging thick biomedical sample remains unclear, as all the experiments introduced additional artificial optical aberration to the imaging system. It raises the question of whether introducing such aberration could be necessary in practical scenarios. The author pointed out that "Correcting for this aberration further improves the polymer-free tomogram, as can be observed in the corrected tomogram, where red blood cells show enhanced axial resolution compared to the polymer-free tomogram.". Nonetheless, a systematic quantification of the improvement over the polymer-free image are essential.

Considering the manuscript's content and focus, it might be a better suit for a more specialized optics journal.

Additional comments:

1. About "target volume" vs "aberrating media":

- a. In the Preface of the reply: "As the sample becomes thicker, the aberration will vary within the target volume, but our method can still determine the average aberration within this volume. Consequently, our aberration model addresses the aberration caused by the aberrating media, rather than the target volume." Stating that the method "can still determine the average aberration within this volume" yet "our aberration model address ... rather than the target volume" appear to be confusing. Moreover, what's the definition of the "target volume" versus "aberrating media"? Is the volume of the tissue sample outside of the target plane considered as the aberrating media or part of the target volume?
- b. What about the two plastics in Figure 8? In Page 8, the author mentioned that correcting of incoming aberration $\$P_in\$$ is not necessary, is it still the case for Figure 8, where the sample is sandwiched by the plastic.

2. Figure 4:

- a. Panel n: confusing legends and missing figure labels
- b. Panel o: This legend is conflicting with the main text and the colors appear to be incorrect. Same as the legend for Figure 5h.

(Remarks on code availability)

Reviewer #2

(Remarks to the Author)

I am convinced by the authors' response and the new results included in the paper (Fig.~8) and the Supplementary Material.

Just one minor comment:

The authors refer to CLASS and the distortion matrix approach as conventional AO methods. This terminology can be

misleading as those methods are computational and not conventional in the field of adaptive optics. They are both based on the post-processing of the reflection/transmission matrix, which is rather new in the field of adaptive optics. I would suggest to refer to those approaches as matrix imaging methods or alternative CAO methods (CAO: computational adaptive optics).

(Remarks on code availability)

Reviewer #4

(Remarks to the Author)

(Remarks on code availability)

Version 2:

Reviewer comments:

Reviewer #1

(Remarks to the Author)

The authors have failed to apply their method to realistic tissue, which is the gold standard in this field. Please see attached PDF for a full report, with illustrations.

(Remarks on code availability)

NA

Reviewer #4

(Remarks to the Author)

(Remarks on code availability)

Version 3:

Reviewer comments:

Reviewer #1

(Remarks to the Author)

We had asked for additional data - taken with the author's set-up - using a biological sample. None was forthcoming. The authors apply their method and an existing method to the published data. There is an improvement using the author's methods relative to the prior "CLASS" method.

(Remarks on code availability)

Reviewer #4

(Remarks to the Author)

(Remarks on code availability)

Preface

We thank the reviewers for their careful reading and helpful comments on validation and technical details that we had overlooked. In response, we have made significant changes to the manuscript that we believe have strengthened it and improved the clarity of our findings. Changes made in the main text and Supplementary Information are highlighted. Given the extensive changes made to the revised manuscript, we begin with a brief summary of the major changes.

Summary of the major changes

1. Clarification of the aberration model

Reviewers have asked about the aberrations that our method can correct. To clarify this, we have added the following illustration.

Figure R1: Comparison of aberration models. An excerpt from Fig. 1 in the revised manuscript.

As depicted in the figure, our aberration model divides the sample into the aberrating media and the target volume. The target volume, where significant scattering occurs, can be set to any desired depth when using the time-gating method (See the main text for more detailed explanation). The key difference between our model and the conventional model is highlighted in Figs. R1b (conventional) and R1c (ours). While the conventional method assumes that the target volume lies on a two-dimensional plane, our model allows it to have a thickness. This relaxation of the assumption enables our method to work in scenarios where the traditional method fails. As the sample becomes thicker, the aberration will vary within the target volume, but our method can still determine the average aberration within this volume. Consequently, our aberration model addresses the aberration caused by the aberrating media, rather than the target volume.

We have also reformulated the theory to make the concepts more easily referenced, specifically describing the discrimination of incoming and outgoing aberrations by factorizing the aberration matrix.

2. Sample-induced aberrations

In the revised manuscript, we placed more emphasis on sample-induced aberrations that vary with location and described how our method can be applied in such situations. To complement the manuscript, we added an experiment demonstrating the correction of aberrations caused by two pieces of plastic sandwiching the sample, as illustrated in Fig. R2 (Fig. 8). Detailed analysis and other results are presented in the main text.

Figure R2: Experiments addressing spatially-varying (sample-induced) aberrations. An excerpt from Fig. 8 in the revised manuscript.

3. Comparison with conventional method

In the revised article, we strengthened our analysis by comparing our method with conventional methods (CLASS or distortion matrix). We confirmed that the conventional method does not work for transmission imaging of the 10- μm thick tissue (Fig. 4). For a systematic comparison, we also simulated time-gated reflectance imaging at a depth of 100 μm . We compared our method with the conventional method and found that our method performs better when the aberrations are strong enough that the PSF is larger than the isoplanatic patch (Last column in Figs. R3 and S2). Detailed analysis and other results are presented in Supplementary Section 1.

Figure R3: Imaging and focusing through simulated tissue model. An excerpt from Fig. S2 in the revised manuscript (Supplementary Information).

Point-by-Point response

Reviewer #1 (Remarks to the Author):

This manuscript introduces a novel digital aberration correction technique based on the angular optical memory effect. By computing the phase differences between the light field of different incident angles, the authors were able to extract the phase information to correct for the aberration from the optical system and an artificially introduced aberrating optical element. Using a transmission-mode holotomography setup, the authors validated their approach experimentally. There are multiple areas where substantial improvements are necessary to meet the publication standards of Nature Communications. The review below outlines specific aspects that, if addressed, could potentially enhance the clarity, rigor, and contribution of this work to the optics community.

We thank the reviewer for the careful review and criticism. We have tried our best to respond to your comments carefully.

Major Concerns:

The work in this manuscript motivated by the need to image structure within optically dense samples, such as biological tissues. However, the primary demonstrations use an “aberrating layer”, consisting of two lenses and a phase object, to introduce a “controlled aberration”. The nature and significance of the aberrations introduced by this layer should be quantified, as the relevance and effectiveness of the author's technique to the spatially continuous aberrations found in biological samples is unclear.

Thank you for suggesting experiments to improve the manuscript. As indicated in the summary above, we have added experiments on spatially varying aberrations to the main text (Fig. 8) and simulations of deep tissue imaging to Supplementary Section 1.

Comparative experiments, like those in the cited literature such as Kang et al 2015 (Nature Photonics), Kim et al 2019 (Nature Communications), and Jo et al 2022 (Science Advance), would substantially strengthen the paper. These prior works demonstrated aberration correction in biological tissue, including imaging beads embedded in thick rat tissue and in vivo imaging, and provides a much clearer demonstration of aberration correction.

We have adopted a quantitative metric for comparison between the corrected and polymer-free measurements in the main text, which we describe in detail in [Comment 1.2]. We also analyzed the performance of different aberration correction methods in the simulation of deep tissue imaging (Supplementary Section 1). Specifically, the performance was quantified by the Strehl ratio, which is defined as the peak intensity of the focus relative to that of the ideal focus.

Other questions and comments:

[Comment 1.1]

1. Mathematical model for aberration and scattering.

a. “To account for the volumetric properties of a sample, we generalize Eq. (3) by introducing a linear operator T to represent the scattering from the sample, rather than: $S(r)$ ”.

In Eqn (4), is r defined on 2- or 3-dimensional space?

We apologize for any confusion caused by mixing the notation of 2D and 3D spaces. In the revised manuscript, all coordinates are presented in 2D (parallel to the image plane) unless explicitly denoted as r_{3D} or k_{3D} .

b. Using the expansion in Eqn 5, the resulting Eqn 8 is independent of the linear operator T . It seems that ultimately the approach described in this paper is reduced to 2-dimensional.

That's right. Since the wave vector of monochromatic light has a fixed length, $k_0^2 = k_x^2 + k_y^2 + k_z^2$, its wavefront can be completely described in 2D k -space.

According to "Finally, the aberration is digitally corrected through the deconvolution of the outgoing fields E_{out} with the retrieved PSF P_{out} ", it seems that the author's approach only corrected P_{out} . If this is indeed the case, the authors should provide experimental evidence showing the advantages of the proposed approach over the traditional method that accounts for both P_{out} and $S(r)$, as stated in the main text (Eqn 3).

We added results from applying the conventional method to a 10- μm thick tissue to Fig. 4, displaying the 2D image $S(r)$ before and after correction. While the intensity of $S(r)$ increased 300-fold after correction, it could not address the aberration, thus failing to reveal the obstructed biological structure of the tissue.

c. Clarity: The detail of the algorithm is split into three parts, some in the main text, some in the method, and some in the supplementary material, making it hard to follow and understand the overall approach. Improving the organization and presentation is needed to make the manuscript more accessible to readers.

We apologize for that. We have integrated the key explanation of our method into the main text, defining the aberration matrix for mathematical clarity. The detailed processing of aberration matrices, including reweighting and factorization, is now solely described in Methods section.

[Comment 1.2]

2. Aberration correction with a 10- μm thick human tissue.

a. "Intriguingly, the aberration's phase accurately reflected the polymer's wavy texture, as shown in Fig. 2b."

Please explain how the phase accurately reflects the polymer's texture.

Phase retardance is related to the optical thickness of the polymer by the equation $\phi = \frac{2\pi}{\lambda}(n - 1)d$, where n is the refractive of the polymer, λ is the wavelength of light used, d is the physical thickness of the polymer. Thus, the depth profile of the polymer is directly integrated into the phase of the outgoing aberration function. We have also updated the manuscript accordingly.

b. According to the text "after polymer removal, un-aberrated fields still exhibit system-induced aberrations...". Thus the "without polymer" images were not corrected for "system-induced aberration". Nominally, we expect the aberration-corrected image to be much sharper than the images taken "without polymer", as the correction should apply to both system aberration and sample-induced aberration. Yet, from the image and according to the author's interpretation, the aberration correction mainly corrects the aberration introduced by the polymer, which

is external to the sample per se.

We apologize for any confusion caused by our previous statement. Our method corrects both sample-induced and system-induced aberrations. Consequently, the corrected tomogram can exhibit higher quality than the tomogram obtained without the polymer (we avoided the use of the word "unaberrated" here). We have clarified this point in the revised manuscript: "System-induced aberration typically arises due to imperfect optical elements and misalignment of the elements and samples. Correcting for this aberration further improves the polymer-free tomogram, as can be observed in the corrected tomogram, where red blood cells (indicated by white arrows) show enhanced axial resolution compared to the polymer-free tomogram."

c. Does applying the system aberration correction obtained in Supplemental Figure 3 to the "without polymer" images here improve the image sharpness?

Yes, that is precisely what we have demonstrated in the Supplementary Information (Section 4 after revision). We increased the tilt angle (Δk) by a factor of 5 to enhance accuracy, considering that system-induced aberrations are typically small.

d. It would be easier to compare the detailed differences between the "corrected" images and the "without polymer" images if the overlay is shown.

e. Figures 3 and 4: More quantitative metrics should be used to quantify the aberration correction. The Fourier spectrum shown as the insert in Figure 5a should also be included. What is the axial resolution? How does it get improved after aberration correction? Can the 3D PSF be estimated?

Reply to the comments d, e:

Firstly, we have added the angular spectrum of each sample to the revised manuscript. Regarding quantitative comparison and 3D PSF or axial resolution, the 3D phase correlation between corrected (or aberrated) fields and polymer-free fields has been introduced (overlaid correlations before and after correction are presented in Figs. 4-6). Mathematical details can be found in Methods section, where it is demonstrated that phase correlation represents the 3D PSF, assuming negligible aberrations in the polymer-free fields. For example, applying our method to the 50- μm thick sample resulted in a 50-fold increase in peak correlation (from 0.0061 to 0.33), with sharper lateral and axial correlations.

Due to the presence of system-induced aberrations in the polymer-free fields, determining absolute resolution or performance from experimental results is challenging. Therefore, we present simulation results in Supplementary Section 1, where the Strehl ratio (ideal PSF is 1) can be determined since we have the ideal wavefront.

f. Figure 3 legend: What does $\Delta u = 1.03 \lambda^{-1}$ mean? Why does it appear to be much larger than the value mentioned in the main text about Figure 5 "The memory effect range, defined by the 1/e correlation threshold, is quantified as a shift $0.024 \lambda^{-1}$ in the incident spatial frequency"? Presumably, Δu is related to the thickness of the sample, so why do Figure 3 and Figure 4 share the same Δu value?

We apologize for any confusion caused by our previous notations. The angle 1.03λ was not the tilt angle discussed in our formulation; rather, it was a randomly chosen angle used to illustrate the fields under oblique illumination.

To prevent further confusion, we have removed this notation from all figures. Additionally, discussions regarding the tilt angle can be found in Supplementary Section 2: “Based on these findings, the tilt angle was determined to be $\Delta k = 2\pi 0.0054 \lambda^{-1}$, and it was fixed throughout the experiments presented in the main text for convenience”.

[Comment 1.3]

3. Aberration correction with a 50- μm thick human tissue.

a. “This variation is attributed to using different polymer regions for each set of experiments (Figs. 3-5), as the polymer was manually positioned and removed for each trial, and the misfocusing of the sample.” What is the “misfocussing of the sample”? Can it be fixed by moving the objectives toward / away from the sample? It seems that the authors did not attribute the differences in the phase gradient to the increased thickness of the sample.

Defocus aberration occurs when the target volume is not centered in the focal plane, as illustrated below.

Figure R4: Defocus aberration. An excerpt from Fig. S6 in the revised manuscript (Supplementary Information).

Defocus aberration is often caused by difficulty in focusing precisely on the center of thick samples by eye. As you mentioned, it can be corrected physically by adjusting the distance between the sample and the objectives. In Supplementary Section 4, we discuss defocus aberrations in more detail.

b. “This variation is attributed to using different polymer regions for each set of experiments (Figs. 3-5), as the polymer was manually positioned and removed for each trial, and the misfocusing of the sample.” Why does the polymer need to be removed after each trial? If the polymer does not need to be removed, do the author expect a largely similar phase gradient when switching from a 10 μm sample to a 50 μm sample?

We displaced the polymer each time to measure the polymer-free fields. We expect that the phase gradients and aberration functions are nearly identical when measuring both samples without touching the polymer, disregarding defocus aberration and tilt aberrations discussed in Supplementary Section 4. Addressing your concerns, the increased thickness of the sample results in a larger decorrelation effect (last term of Eq. 7), which can also be interpreted as (axial) variance in the PSF within the sample volume. However, as our method averages aberrations within the target volume, we observed that this thickness effect remains manageable up to 50- μm thickness, as evidenced by the highest peak correlation observed in the 50- μm thick sample. The decorrelation effects become evident in the 100- μm thick sample, which exhibits the lowest peak correlation: “This limitation is depicted in

Fig. 6c, where the 3D correlation of the corrected fields shows lateral confinement along with axial elongation due to the depth-dependent nature of aberrations.”

c. Figure 4f: It would be clearer if the author could outline the position of the vessels and fibers.

We have added annotations to Fig. 5 (now renumbered) in the revised manuscript.

[Comment 1.4]

4. Memory effect in a 100 μm thick human tissue and its influence on aberration correction

a. Figure 5b also suggests that the proposed technique corrects aberration induced by the polymer but not in the biological sample.

As discussed in the above summary of changes: “As the sample becomes thicker, the aberration will vary within the target volume, but our method can still determine the average aberration within this volume. Consequently, our aberration model addresses the aberration caused by the aberrating media, rather than the target volume.”

We considered the thickness of the target volume to broaden the applicability of our method to situations where conventional methods may fail. Conversely, our method can address sample-induced aberrations caused by surrounding tissue layers in scenarios where conventional methods are applicable. For example, aberration correction methods are commonly employed in time-gated reflection imaging (OCT), where the target volume and depth are determined by the resolution of time gating and the reference mirror, respectively. To prove this, we have included simulation results in Supplementary Section 1, which demonstrate the advantages of our method over conventional approaches in the presence of strong aberrations.

Furthermore, we have extensively discussed the concept of the target volume in the revised manuscript.

[Comment 1.5]

5. Aberration correction with moving samples:

a. “Our aberration correction method is adept at handling samples in motion, as long as their movement is slower than one-third of the camera's frame rate”. The unit of movement should be [Length / Time] instead of [1/Time]. Additional discussion/analysis on the speed of the sample movement should be added.

We are grateful for the comments that enhance the quality of our manuscript. In the revised manuscript, we have included the following discussion: “Therefore, motion has little effect on the performance if the decorrelation time of the scattered fields is longer than the time interval between the consecutive captures. In terms of the velocity, our method is effective for samples that move less than the diffraction limit during the interval. For example, our setup allows for movement up to 30 $\mu\text{m/s}$ given the 60 Hz frame rate of the camera.”

b. “By measuring the components of the gradient separately, we can further extend our method's applicability to dynamics approaching half the camera's frame rate.” How to “measure the components of the gradient separately”? How does it differ from using the second measurement as the reference instead of the first one?

Thank you for pointing that out. As you mentioned, the aberration matrices can be constructed using only adjacent

frames, as long as the decorrelation time is longer than the time between captures. For example, the columns of two aberration matrices $A_{\Delta k=k_1-k_2}$, $A_{\Delta k=k_2-k_3}$ can be derived from three successively measured frames: $E(k_{in} = k_1)$, $E(k_{in} = k_2)$, and $E(k_{in} = k_3)$. We have updated the manuscript accordingly.

c. “After correcting for aberration, the fields accurately represent the beads' shape, facilitating the analysis of bead dynamics at frequencies up to 20 Hz”: Why not at the full 60 Hz frame rate? Is the system actively correcting for the aberration using every 3 measurements?

Thank you for your observation. We have identified the sentence in question as erroneous and have consequently removed it from the manuscript.

[Comment 1.6]

6. Discussion and conclusion

a. “The strategic utilization of slight tilts in incidence angles within the angular memory effect's range allows for more robust aberration measurement than existing techniques”. The author should provide quantitative evidence/comparison experiments to justify this claim.

As previously stated, a comparative analysis between our method and the conventional methods has been incorporated into the main text and Supplementary Section 1.

[Comment 1.7]

7. Supplemental material section 2:

a. “In Methods section, we introduced a technique for directly integrating the phase gradients of the aberration. This method might necessitate phase unwrapping, which can pose challenges, particularly when working with masked phases. To address this issue, we can utilize the complex exponential of the phase instead of the phase itself.” The shared in “demo_code.m” implemented the method described in the supplemental material. It is not clear whether all the figures and experimental results shown in the main text were based on the algorithm in the supplemental material or in the main text (and the Method section). If they are based on the algorithm in the supplemental material, it should be clarified and integrated into the main text. Otherwise, the author should evaluate the difference in aberration correction from both algorithms.

We apologize for any inconvenience. Throughout the manuscript, we consistently applied the same method, and the corresponding explanation has now been integrated into the subsection ‘Factorization of Aberration Matrix’ in the Methods section.

[Comment 1.8]

8. Supplemental Figure 3: This figure and section are about correcting system-induced aberrations. So

a. Why a 10 μm thick human sample is used for correcting “system-induced aberration”? Why not use fluorescence beads for such system aberration correction?

b. Are the authors assuming that the aberration introduced by the 10 μm thick human sample is negligible? If yes,

then why do the resulting “phase of the aberration in k-space” and the “corresponding PSF” shown in Supplementary Figure SF3a,b look so different? Especially for SF3a, there appear to be complex detailed structures in the PSF. Is this expected? Are FOV 1 and 2 just taken at different positions of the sample?

Reply to the comments a, b:

As previously stated in [Comment 1.3.b], our method demonstrated a limited decorrelation effect due to the thickness of the sample, up to 50 μm . The decorrelation effect can be particularly averaged out as we measure fields with a large number of incident angles. This is why we used the 10- μm thick tissue to investigate sample-induced aberrations. We obtained the aberrations while moving the sample to random positions. Although the aberrations obtained at different positions appeared differently, they became clear when decomposed into defocus aberrations and remaining aberrations (denoted as system in Figs. R5 and S7): “The defocus aberrations appear differently in the two positions, implying difficulty in focusing on the center of thick samples by eye. On the other hand, the system aberrations appear as a skewed doughnut shape in both positions. The doughnut shape is indicative of spherical aberrations, while the skewness indicates coma due to the tilt of the sample. While the shape of the system aberrations remained almost the same, they differed in the skew direction, suggesting a change in the tilt direction when the sample was moved.”

The tilt aberration is frequently encountered, particularly when using water-immersion objectives (see Arimoto et al. *J Microsc.* (2004) doi: 10.1111/j.0022-2720.2004.01383.x.).

Figure R5: Aberrations measured at different positions of sample. An excerpt from Fig. S7 in the revised manuscript (Supplementary Information).

[Comment 1.9]

9. The authors indicated that the main text includes “a complete, detailed description of the code’s functionality (i.e. pseudocode)” in the “Code and Software Submission Checklist”. However, it seems that neither the pseudocode nor a summarizing description of the entire algorithm could be found in the main text.

The code is comprised of two main parts: (i) the construction of the aberration matrix using Eq. (6), and (ii) the reweighting and factorization of the matrix according to the corresponding sections in Methods. While we believe that the revised manuscript provides clear descriptions of the code, we have made the code openly available, including the simulation of time-gated reflection as discussed in Supplementary Section 1, to facilitate the reproduction of our work.

[Comment 1.10]

10. The authors also provided MATLAB implementation and test data, which helps the reader to better understand their approach. However, the author only provided part of the code. As a manuscript focused on novel methodology, providing the dataset and the code for replicating the figures is essential. Regarding the “demo_code.m” in the provided Google Drive link:

a. The tilted angles are $[-0.8857, 0.1133]$ and $[0.1167, 0.8590]$. How were these tilted angles determined? They are not exactly orthogonal (dot product -0.006). Is this a numerical error? Alignment error?

The tilt angles were experimentally calibrated to compensate for the alignment error of the optical elements. The error was compensated so that the two tilt directions are orthogonal to each other. However, it is possible that residual error may exist, as you pointed out.

b. The ``mk_ellipse`` function is missing, without which the demo code cannot be run.

We apologize for any inconvenience. As mentioned in the Code and Data Availability section, we have now uploaded our codes and data to GitHub. Please let us know if you encounter any further issues.

c. What is ``noise`` and how is its value 200000 determined? If other laboratories want to replicate the measurement and technique, how should they determine this value?

It specifies the threshold value used for the aberration matrix. We have provided details on how we determined this threshold in the revised manuscript: “In the case of transmission imaging, the high scattering anisotropy factor of biological tissues results in a narrow angular spectrum of transmitted light. Consequently, the diagonal elements are much more heavily weighted than the off-diagonal elements, leading to ineffective factorization due to the high matrix rank. A better strategy is to threshold the matrix elements above the noise floor, rather than simply utilizing the modulus as the weight. We manually adjusted the threshold so that the argument of the matrix exhibits smooth phase gradients, thereby removing impulse noise. (see Supplementary Section 2).”

The figure below visualizes how the noise in the argument of the aberration matrix is significantly reduced after thresholding.

Figure R6: Thresholding of the aberration matrix.

To facilitate the reproduction of our work, we have also made the code openly available.

d. What is `mask_size` and how was its value 0.2 determined? How does it affect the aberration correction result?

We masked the aberration matrix to filter out the peaky noise from the DMD. We have elaborated on this in the revised manuscript: “In our experimental setup, the DMD notably acts as a significant source of the noise, producing multiple peaks in k-space as shown in Supplementary section 5. Since these artifacts can exceed the threshold when the angular spectrum is narrow, we additionally mask the matrix elements far from the diagonal before factorization. For samples with lower scattering, such as in the 10- and 50- μm tissues, we limit the masking radii to 0.3–0.4 NA. In contrast, masking is not required for the 100- μm thick tissue sample.”

Conclusion:

This manuscript addresses a topic of interest to the field of optics. Yet from the image and according to the author’s interpretation, it seems that aberration correction mainly corrects the aberration introduced by the polymer, which is external to the sample per se. This is a fundamental weakness of the current manuscript.

We trust that the revised manuscript adequately addresses all your concerns, particularly those regarding sample-induced aberrations, which are newly addressed in Fig. 8 and Supplementary Section 1.

Reviewer #2 (Remarks to the Author):

This paper deals with the compensation of aberrations in a transmission-mode holotomography setup. Exploiting a tilt-tilt memory effect, the authors are able to compensate for aberrations induced by a polymer layer in a plane conjugated with the pupil plane of the microscope objective in transmission. More precisely, the method consists in extracting the phase derivative of the aberration phase law by comparing the wave-fields transmitted by the sample and aberrating layer when slightly tilting the direction of the incident plane wave. The choice of this tilt is subject to the following compromise: large enough such that noise does not hamper the numerical integration of the aberration phase law; small enough to benefit from the memory effect that scales as λ/L , with L the thickness of the sample. Based on this method, the authors manage to unscramble the aberrations induced by the polymer layer and retrieve a complex field close to the reference wave-field measured without the coating layer. The compensation of aberrations is also illustrated by building refractive index tomograms using gradient based Rytov approximation.

We thank the reviewer for the summary and criticism. We have made every effort to respond to your comments carefully.

Although the method is technically interesting and appealing, its range of application seems restricted to the compensation of optical system aberrations. Yet, the real problem for deep imaging in tissues is not the aberrations of the imaging system for which a wealth of correction methods already exists but the correction of sample-induced aberrations. Such aberrations cannot be compensated by the method proposed by the authors, or at least they do not prove it and even not discuss about this possibility. In that view, the abstract and introduction of the paper are misleading since the authors claim that their method “enhances imaging of thick human tissues” (quoted from the abstract). A correct statement would be that the method allows the compensation of far-field aberrations induced by a thin phase screen. For these reasons, I cannot support publication in Nature Communications for the paper in its current form. However, as I said, the proposed method is interesting and could deserve publication in more specific journal.

We agree with you that the previous manuscript lacked the demonstration of sample-induced aberrations. We have strengthened the narrative about this subject in the revised manuscript. In particular, we have added a new section both in the main text (Fig. 8) and Supplementary Information (Figs. S1-3). The main text now includes a correction of aberrations caused by two pieces of plastic sandwiching the sample. In Supplementary Section 1, we provide a comparative analysis of different correction methods in simulated time-gated reflection imaging, demonstrating how aberrations induced by the upper tissue layers can be corrected. A detailed explanation is provided in the response to the comment below.

Please find below more detailed comments:

[Comment 2.1]

1/ One major flaw of the paper is the confusion about the “thick tissue” application that can let the inexperienced

reader think that this method is dedicated to the compensation of sample-induced aberrations and multiple scattering. I suggest the authors to be more modest about the range of application of this work. The real challenge for deep imaging in transmission is solving the inverse problem that links the transmitted wave-field with the refractive index distribution as mentioned briefly by the authors in the conclusion (page 15, line 3) :“though challenges in tomogram reconstruction at this thickness highlight areas for future improvement.”

Sample-induced aberrations are characterized by their position-dependent nature. Therefore, methods such as windowing or time-gating are applied to obtain the aberration at a specific location. However, in the presence of scattering or strong aberrations, these methods cannot determine a single aberration, and the existing methods show problems.

As discussed in the above summary of changes: “As the sample becomes thicker, the aberration will vary within the target volume, but our method can still determine the average aberration within this volume. Consequently, our aberration model addresses the aberration caused by the aberrating media, rather than the target volume.”

We considered the thickness of the target volume to broaden the applicability of our method to situations where conventional methods may fail. Conversely, our method can address sample-induced aberrations caused by surrounding tissue layers in scenarios where conventional methods are applicable. For example, aberration correction methods are commonly employed in time-gated reflection imaging (OCT), where the target volume and depth are determined by the resolution of time gating and the reference mirror, respectively. To prove this, we have included simulation results in Supplementary Section 1, which demonstrate the advantages of our method over conventional approaches in the presence of strong aberrations in the multiple scattering regime (the target depth is set at twice the mean scattering path.).

In addition, our method can also address aberrations induced by weakly scattering aberrating media shown in Figs. R2a and 8a: “Since the scattering inside the plastic pieces is more anisotropic than inside the tissue, the plastic pieces and tissue can be distinguished as aberrating media and target volume, respectively, after windowing the fields.” (p.15, main text)

[Comment 2.2]

2/ Just before, the authors claim: “This approach surpasses traditional methods in handling thick tissue samples”. This assertion shall be supported by a quantitative comparison with traditional methods. For instance, a comparison with CLASS algorithm that is based on the same idea (tilt-tilt memory effect) would be relevant. As they measured the transmission matrix across the sample, the correlation methods that the authors mention in the introduction could be also implemented in post-processing for comparison purpose.

Thank you for your advice on improving the manuscript. In the revised article, we strengthened our analysis by comparing our method with conventional methods (CLASS or distortion matrix). We confirmed that the conventional method does not work for transmission imaging of the 10- μm thick tissue (Fig. 4).

For a systematic comparison, we also simulated time-gated reflectance imaging at a depth of 100 μm . Specifically, the performance was quantified using the Strehl ratio, defined as the peak intensity of the focus relative to that of the ideal focus (Figs. R7g and S2g). From the comparative analysis, it was found that our method performs better

when the aberrations are strong enough that the PSF is larger than the isoplanatic patch (last column in Fig. R7).

Figure R7: Comparative analysis in imaging and focusing through simulated tissue model. An excerpt from Fig. S2 in the revised manuscript (Supplementary Information).

[Comment 2.3]

3/ To be more quantitative, could the authors use a metric that will allow to compare the image they obtain with the ground truth image obtained without the polymer layer? For unexperienced eyes, it is difficult to assess the image quality in the different cases shown by the authors.

Regarding the quantitative metric, we have introduced the 3D phase correlation between corrected (or aberrated) fields and polymer-free fields (Figs. 4-6). Mathematical details can be found in Methods section, where it is demonstrated that phase correlation represents the 3D PSF, assuming negligible aberrations in the polymer-free fields. For example, applying our method to the 50- μm thick sample resulted in a 50-fold increase in peak correlation (from 0.0061 to 0.33), with sharper lateral and axial correlations.

Due to the presence of system-induced aberrations in the polymer-free fields, determining absolute performance (or Strehl ratio) from experimental results is challenging. Therefore, we present simulation results in Supplementary Section 1, where the Strehl ratio (ideal PSF is 1) can be determined since we know the ideal wavefront.

[Comment 2.4]

4/ The authors fail in showing in which practical situations this method can be useful. For instance, in the present case, the method addresses the aberrations induced by the polymer layer included on purpose in a very specific plane of the imaging system (output pupil plane of the MO). However, it does not seem to compensate for spherical aberrations induced by the imaging system itself. For instance, for the results shown in Fig.4, the authors had to refocus their tomogram by $+3.5 \mu\text{m}$ to compensate for defocus aberration. Hence it means, that their approach is not able to compensate for this basic aberration that seems to be quite common in the field. Hence, the question is: For which kind of aberrations the method is good for?

As discussed in the above summary of changes, our aberration models detects aberrations arising from aberrating media, as well as defocus and system-induced aberrations. In the revised Supplementary Section 4, we discuss defocus and system-induced aberrations in detail. Defocus aberration (which was identified as $+3.5 \mu\text{m}$ in the experiment you mentioned) occurs when the target volume is not centered in the focal plane (Figs. R8a and S6a). In a time-gated microscope, such as optical coherence tomography (OCT), the defocus aberration can be characterized by the offset between the focal plane and the reference mirror (Figs. R8b and S6b).

Figure R8: Defocus aberrations denoted by black arrows. An excerpt from Fig. S6 in the revised manuscript (Supplementary Information).

In our experiments, defocus aberrations are caused by difficulty in focusing precisely on the center of thick samples by eye. On the other hand, system-induced aberrations, such as spherical aberrations, arise due to imperfections in optical components. Our method can correct these system-induced aberrations, as demonstrated by applying our method after displacing the polymer layer (Fig. S7).

[Comment 2.5]

5/ Related to that questions: If the aberrating layer is placed between the microscope objective and the sample, can we expect the method to work to some extent? Or is the method restricted to pupil aberrations?

7/ One interesting feature of the proposed method is the ability to discriminate between input and output

aberrations. However, the paper does not demonstrate this feature since the polymer layer is only included in the output arm.

Reply to the comments 5/ and 7/:

Thank you for suggesting experiments to improve the manuscript. Our method is applicable in the situations you mentioned. As indicated in the summary above, we have added experiments on spatially varying aberrations to the main text (Fig. 8), demonstrating the correction of aberrations caused by two pieces of plastic above and below the sample.

Figure R9: Aberrations induced by two pieces of plastic above and below the sample. An excerpt from Fig. 8 in the revised manuscript.

[Comment 2.6]

6/ The authors said that their method is robust with respect to motion of the sample. I guess this is not the case for time-varying aberrations, it may be fair to specify it.

We agree with your comment. We have revised the manuscript as follows: “Aside from movement within the target volume, the aberrations should remain unchanged throughout the total measurement. Nevertheless, we expect the time scale of aberration changes to be much larger than that of the target, since the isoplanatic patch is typically larger than the diffraction limit.”

[Comment 2.7]

Besides these questions about the scope of this work, please find below some more specific points:

8/ Page 5, line 16: “It is worth noting that the linear operator T is equivalent to the scattering matrix of a sample, which comprises the transmission matrix and reflection matrix^{38,39}.”

The linear operator is equivalent to the transmission matrix, not to the whole scattering matrix.

The sentence has been modified as follows: “It is worth noting that the linear operator T is equivalent to the reflection or transmission matrix of a sample when imaging reflected or transmitted light, respectively”

[Comment 2.8]

9/ I have some doubt about the way the memory effect is included in the analytical model presented by the authors. There is no impact on the method but it may be misleading for readers. In Eq. 5, the angular memory effect is

described as follows:

$$T_{(k_{in}+\Delta k)}(k+\Delta k)=T_{(k_{in})}(k)+O(\Delta k/k_{MER})$$

This expression suggests that the transmitted wave-field increases with Δk , which is non-sense. A normalization term is needed to avoid such inconsistency. According to me, a proper way to include the memory effect would be to introduce the correlation function, $C(\Delta k/k_{MER})$, characteristic of the memory effect between $T_{(k_{in}+\Delta k)}(k+\Delta k)$ and $T_{(k_{in})}(k)$, such that

$$C(\Delta k/k_{MER})=\langle T_{(k_{in}+\Delta k)}(k+\Delta k) T_{(k_{in})}^*(k) \rangle$$

The bias made on the estimation of the aberration law could be derived by considering this correlation function, which is actually plotted in Fig.5.

Thank you for pointing that out. We have modified Eq. (3, renumbered) as you suggested.

[Comment 2.9]

10/ There is a problem with equation numbers in the Methods section. The authors are referring to an equation number that does not correspond to the equation they want to refer to. For instance, page 27, line 3, they refer to Eq.14 to demonstrate Eq.14. The same problem appears further. Please double check all the equation references.

We apologize for any inconvenience. We thoroughly checked the equation and figure numbers during the manuscript revision process.

Reviewer #3 (Remarks to the Author):

Dear Editor and authors,

Correcting optical aberrations in a thick sample is challenging. Typically, it requires to insert a “guide star” within the sample or using an iterative wavefront correction. Those solutions are not appropriate for all samples due to the temporal decorrelation or due to the impossibility to insert non-invasively a guide star. In this article, the authors proposed and demonstrate a new approach to measure aberrated wavefronts that doesn’t require a guide star or a long iterative feedback correction in a holographic microscope configuration. Thanks to the angular memory effect of the sample, illuminating a sample by a couple of planes waves separated by a small angle leads to the measurement of the derivative of the aberrated wavefront. The wavefront obtained then is used to digitally correct holographic images. The authors applied their approach in two proof-of-concept scenarios: for 3D refractive index imaging and holographic imaging of dynamic samples, where a strongly aberrated layer is added after the sample.

The concept is elegant, and the proof-of-concept experiments are convincing. But the applicability seems limited because some theoretical derivations aren’t clear and because the limitations of the techniques aren’t much discussed. Some additional points need to be clarified.

We thank the reviewer for the careful review and criticism. We have refined the derivation, particularly by defining the aberration matrix for mathematical clarity.

Regarding limitations, we have added the following discussion in the Conclusion section: “The limitations of our method are consistent with those of gradient-based wavefront sensing methods. Integrating the phase gradients converts white noise into Brownian noise, increasing the error in lower-order aberrations. To improve accuracy, errors in the phase gradients should be sufficiently suppressed, requiring careful consideration of the number of incident angles and the magnitude of the tilt angle. In addition, we used an inverse gradient algorithm based on the least-square method, which neglects branch points (phase singularities) that generally appear in sample-induced aberrations in the multiple scattering regime. To address this discrepancy, the phase of the branch points needs to be obtained separately”

Detailed responses to your comments are provided below.

[Comment 3.1]

At first, I’m a bit confused with the notations used by the authors. In equation (4), T is defined as an equivalent to the scattering matrix of the sample. But if T is a complex matrix, what allow the authors to go from a matrix product to an Hadamard product? I feel an assumption is missing here that will help the reader to understand the limitation of this approach.

This is based on the convolution theorem, $\int P(r_1) * T(r_1, r_2) dr = P(k_1)T(k_1, r_2)$, which states that the convolution operation $[P(r_1) *]$ becomes an element-wise multiplication in the k -space representation. Therefore, the effect of the aberration can be described by the Hadamard product by changing the basis from the

position space to k -space (Eq. (4)). In other words, the transmission matrix of aberrating media appears as a diagonal matrix in k -space within an isoplanatic patch.

[Comment 3.2]

I feel there is a Δk range where the wavefronts gradients can be successfully reconstructed (whatever the angular memory effect of the sample). If Δk is too small, the sensitivity might be too low to measure the gradient in presence of noise. If it's too large the derivative might be misestimate. Can the authors precise/quantify the Δk range validity of their approach?

As you noted, the appropriate choice of the tilt angle (Δk) is critical to our method because the magnitudes of both the signal and error components are affected by the tilt angle. In the revised Supplementary Section 2, we have summarized the magnitudes of these components in Table R1 (Table S1).

Type	Component	Magnitude
Signal	Gradient of an aberration function	$\mathcal{O}(\Delta k)$
Error	Decorrelation term	$\mathcal{O}(\Delta k^2/k_{\text{MER}}^2)$
	Measurement noise	$\mathcal{O}(T ^{-2})$

Table R1: Signal and error components in the phase of an aberration matrix.

It can be observed that the magnitudes of both the signal and error components depend on the tilt angle. Consequently, the signal magnitude can be maximized by adjusting the tilt angle. To reduce the effect of the decorrelation term, it is necessary to keep the tilt angle below the memory effect range. At the same time, the tilt angle should be sufficiently large so that the desired signal is greater than the noise.

Figure R9: Aberration matrices with different tilt angles. An excerpt from Fig. S4 in the revised manuscript. (Supplementary Information).

We further investigated this issue by directly observing the phase of the aberration matrix (Figs. R9 and S4): “As

the tilt angle increases, the signal, which is the gradient of the outgoing aberration function, becomes stronger. This effect is particularly notable in the polymer-added case compared to the other cases. At the same time, the phase become increasingly randomized, resulting in a diminished correlation, as illustrated in Fig. 6d.”

From this observation, we conclude that the upper bound of the tilt angle is approximately the memory effect range ($\Delta k = 2\pi \cdot 0.022/\lambda$, third column in Fig. R9), while the lower bound is determined by the strength of the measurement noise. Although there is no strict limit where the method suddenly fails, these bounds provide practical guidance for choosing an appropriate tilt angle.

[Comment 3.3]

To go further, this approach requires some angular memory effect. How much? The authors touched this issue with the 100 μm thick sample in figure 5. But it would be interesting to understand better. Can the authors link the memory effect range needed to the Δk range validity of their approach to estimate the tissue thickness they will be able to image?

The increased thickness of the sample results in a larger decorrelation effect mentioned in Comment 3.2, which can also be interpreted as (axial) variance in the PSF within the sample volume. We observed that this thickness effect remains manageable up to 50- μm thickness, as evidenced by the highest peak correlation observed in the 50- μm thick sample. The decorrelation effects become evident in the 100- μm thick sample, which exhibits the lowest peak correlation: “This limitation is depicted in Fig. 6c, where the 3D correlation of the corrected fields shows lateral confinement along with axial elongation due to the depth-dependent nature of aberrations.” In summary, increased thickness reduces the memory effect range, thus impairing accuracy. While it is difficult to specify an exact thickness limit as it depends on the desired level of accuracy, we believe that a correction of up to 50 μm thickness is practically sufficient. This is because the effective thickness of the sample can be reduced in time-gated reflection imaging employing a broadband light source, which is typically used for deep tissue imaging.

[Comment 3.4]

To distinguish between incident and outgoing aberrations, the authors assumed that the incident aberrations are the same for different incident wave vectors. It’s equivalent to assume that the incident aberrations are only due to the optical system. Is it always true? In a realistic scenario, where the aberrations to correct are induced by the sample itself, both incident and outgoing aberrations will depend on the incident wave vectors. Will the approach suggested by the authors still work?

Our method can handle aberrations for both incoming and outgoing light, as demonstrated in the correction of spatially varying aberrations caused by two pieces of plastic above and below the sample (Figs. R2 and 8). We apologize for any misleading statements and derivations in the previous manuscript. We have revised the derivation by defining the aberration matrix, which is visualized in Figs. R10 and 2. The aberration matrix can be constructed from the measured phase (Eq. (6), Fig. R10b.ii), which exhibits the aberrations for outgoing and incident light in its columns and rows, respectively. Our previous statement, which was found to be misleading, was intended to indicate that a column of the aberration matrix is affected by a constant phase from the incoming

aberration. Now that we have introduced the concept of the aberration matrix, we believe that the process of separating outgoing and incoming aberrations is more clearly described as a matrix factorization problem.

Figure R10: Revised schematic of the aberration detection process. An excerpt from Fig. 2 in the revised manuscript.

[Comment 3.5]

Some phrasings are misleading. I quote from the Discussion and Conclusion: “It achieves effective aberration correction in samples up to 100 μm thick”, the authors didn’t show these experiments. They only show an aberration measurement through a 100 μm tissue with an indirect validation.

Thank you for commenting and helping us to improve. In the revised manuscript, we have introduced the 3D phase correlation between corrected (or aberrated) fields and polymer-free fields (Figs. 4-6), which essentially represents the 3D PSF. The added results show an increase in peak correlation and sharpening of the PSF after applying our method to the 100- μm thick sample (Figs. R11c, 6c).

Figure R11: 3D correlations (PSFs) before and after correction. An excerpt from Fig. 6 in the revised manuscript.

[Comment 3.6]

Additionally, the optical configuration used to image the dynamic samples isn’t specified (figure 6). I believe it wasn’t an holotomography measurement as the dynamic of the sample should have prevented to obtain the 76 images for 3D reconstruction in a good condition.

We used the same setup for all the experiments shown in the manuscript. With this setup, we measured holographic images (fields) of transmitted light in response to incident plane waves at different angles, which is equivalent to measuring a transmission matrix. Although the tomogram could technically be reconstructed, we chose not to show it to avoid confusion between the effects of motion blur and aberration.

Minor comments:

[Comment 3.7]

- To justify, I quote “Furthermore, the corrected tomogram showcased enhanced axial resolution of red blood cells, as indicated by white arrows in the xz section, surpassing that of the tomogram derived from unaberrated fields.”, the authors might want to add cross section of the tomograms to illustrate.

In the revised Fig. 5, we present x-y and x-z cross-sections of tomograms.

Figure R12: Aberration correction with 50- μ m thick human tissue. An excerpt from Fig. 5 in the revised manuscript.

[Comment 3.8]

- In figure 5, how is define the field correlation? Is it the norm of $C(\Delta k)$?

Thank you for pointing that out. Yes, it is the norm of the correlation, $|C(\Delta k)|$. We have clarified this in the figure caption.

[Comment 3.9]

- In SI “Tomographic reconstruction using gradient based Rytov approximation”, the equation number on the text is off by 1. For example, in the sentence “In tomographic reconstruction, Eq. (17) is inverted to derive the scattering potential from the ...” I believe we should read Eq. (16) instead.

We apologize for any inconvenience. We thoroughly checked the equation and figure numbers during the manuscript revision process.

Reviewer #4 (Remarks to the Author):

We thank the reviewer for the review.

Reviewer #1 (Remarks to the Author):

The authors have significantly improved the clarity and organization of the manuscript. This work represents an advance in computational adaptive optics. However, its practical value for imaging thick biomedical sample remains unclear, as all the experiments introduced additional artificial optical aberration to the imaging system. It raises the question of whether introducing such aberration could be necessary in practical scenarios.

We appreciate the reviewer's concern regarding the practical applicability of our method. The use of artificial aberrations in our experiments was a deliberate choice to validate the robustness and flexibility of our approach under controlled yet realistic conditions. This is a widely accepted practice in adaptive optics, as it enables systematic evaluation and comparison of novel techniques.

While we acknowledge that more explicit real-world examples could further provide additional clarity, our goal was to maintain the broad applicability of the method by avoiding assumptions about specific forms of aberrations or sample types. This approach ensures that our method is generalizable and relevant across a variety of commonly encountered imaging contexts, such as reflection imaging of biological specimens. Additionally, introducing artificial aberrations for validation has been well-established in the field as a means of method validation. For instance, techniques such as CLASS^{1,2} employed polymer layers as artificial aberrating layers, and similar approaches are widely accepted for testing novel methods.

Moreover, we used biological samples rather than simplified resolution targets to create a more biologically relevant and challenging environment. These samples exhibit lower scattering and contrast, which better simulate real-world imaging conditions, demonstrating the practical effectiveness of our method for biomedical applications.

To further substantiate our claims of practical relevance, we have included detailed simulations of deep tissue reflection imaging in Supplementary Section 1. These simulations show that our method not only generalizes well but also outperforms existing computational techniques in realistic scenarios. This highlights the significant potential of our approach for real-world biomedical imaging applications.

The author pointed out that "Correcting for this aberration further improves the polymer-free tomogram, as can be observed in the corrected tomogram, where red blood cells show enhanced axial resolution compared to the polymer-free tomogram.". Nonetheless, a systematic quantification of the improvement over the polymer-free image are essential.

We appreciate the reviewer's emphasis on the importance of systematic quantification. To address this, we have incorporated a more rigorous quantitative analysis of the improvement over the polymer-free image by introducing the variance of the refractive index as a sharpness metric. This metric, shown in the revised Fig. 5f, provides a reliable means of evaluating image clarity, as a larger variance typically correlates with reduced blurring and enhanced detail in the image.

Figure RR1: Variance as sharpness metric. An excerpt from Fig. 5 in the revised manuscript.

Specifically, the corrected tomogram shows a 3.7-fold increase in variance when compared to the polymer-added tomogram, and a 1.3-fold increase compared to the polymer-free tomogram. These results clearly demonstrate the significant improvement in image sharpness achieved through aberration correction, particularly in the vicinity of the red blood cells. This systematic quantification confirms the enhanced axial resolution and overall image quality, thereby validating the effectiveness of our method.

Considering the manuscript's content and focus, it might be a better suit for a more specialized optics journal.

While we acknowledge the reviewer's perspective on the focus of our work, we respectfully believe that the significance of our findings and their broad relevance to imaging under real-world conditions make this manuscript well-suited for this journal. Our work addresses critical challenges posed by severe aberrations and dynamic environments, which are not only key issues in the field of computational adaptive optics but are also of paramount importance in broader biomedical imaging applications.

Furthermore, the contributions of this study align with the level of impact seen in recent research published in this journal, particularly in addressing advanced imaging techniques under complex, real-world conditions^{3,4}. By advancing both the theoretical and practical aspects of adaptive optics, our work offers new insights and solutions that we believe will resonate with the journal's diverse readership, spanning optics, biomedical imaging, and computational imaging fields.

Additional comments:

1. About "target volume" vs "aberrating media":

a. In the Preface of the reply: "As the sample becomes thicker, the aberration will vary within the target volume, but our method can still determine the average aberration within this volume. Consequently, our aberration model addresses the aberration caused by the aberrating media, rather than the target volume." Stating that the method "can still determine the average aberration within this volume" yet "our aberration model address ... rather than the target volume" appear to be confusing. Moreover, what's the definition of the "target volume" versus "aberrating media"?

Since point spread functions (PSFs) generally vary across a sample, it is crucial to specify a particular location to probe before determining the PSF. In guide star-based adaptive optics (AO), PSFs at specific locations are derived by analyzing light from a chosen guide star. For guide-star-free AO methods, the probed location is restricted both laterally and axially by cropping the measured images and applying a time-gating technique (akin to the depth-sectioning mechanism of optical coherence tomography). We define this restricted region as the target volume.

Existing computational AO methods typically assume that the PSF remains constant within this target volume, referred to as the isoplanatic patch. However, in practice, it is difficult to completely isolate a volume where the PSF is uniform. To account for this, we introduced the concept of the target volume, acknowledging that the PSF may vary to some extent within it.

We recognize that our previous statement, “our aberration model addresses ... rather than the target volume,” may have been unclear. A more accurate description is that our method calculates an averaged PSF within the target volume, even when variations in the PSF occur.

As aberrations become more severe, the increasing size of the PSF makes it harder to isolate an isoplanatic patch, leading to unstable convergence or failure in existing computational AO methods. However, our method can analytically recover averaged aberration matrices within the target volume, even when the PSF varies, by leveraging the memory effect. This capability is demonstrated in our transmission-mode imaging experiments, where time-gating does not provide depth-sectioning functionality.

Is the volume of the tissue sample outside of the target plane considered as the aberrating media or part of the target volume?

The thickness of the target volume is generally related to the depth-sectioning achieved through time-gating techniques. In the absence of time gating, the entire volume containing significant scattering must be incorporated into the target volume. For instance, in the transmission-mode imaging described in the main text, the entire thickness of the tissue becomes the target volume.

When time-gating is employed, as in methods such as CLASS or distortion matrix approaches, the depth of the target volume is determined by the position of the reference mirror, while the thickness is governed by the axial resolution of the time-gating, which typically measures in the tens of micrometers. In the time-gated reflection imaging discussed in Supplementary Section 1, the target volume represents only a portion of the entire tissue—specifically, an 11 μm thickness at a 100 μm depth.

In our framework, the aberrating medium is defined as the region outside the target volume.

b. What about the two plastics in Figure 8? In Page 8, the author mentioned that correcting of incoming aberration $\$P_{in}\$$ is not necessary, is it still the case for Figure 8, where the sample is sandwiched by the plastic.

It is crucial to consider both incoming and outgoing aberrations when estimating aberrations accurately. However, the role of incoming aberrations may vary depending on the measurement basis. In some instances, incoming aberrations may not need to be incorporated into the subsequent aberration correction process. In our particular

case, where the incoming light is composed of plane waves, the incoming aberration solely affects the global phase of the measured images, which does not impact the tomographic reconstruction.

2. Figure 4:

a. Panel n: confusing legends and missing figure labels

b. Panel o: This legend is conflicting with the main text and the colors appear to be incorrect. Same as the legend for Figure 5h.

Thank you for pointing this out. We have updated the manuscript accordingly.

References

- 1 Kang, S. et al. High-resolution adaptive optical imaging within thick scattering media using closed-loop accumulation of single scattering. *Nature Communications* 8, 2157 (2017). <https://doi.org/10.1038/s41467-017-02117-8>
- 2 Moon, J. et al. Second-harmonic generation microscopy with synthetic aperture and computational adaptive optics. *Optica* 11, 128-136 (2024). <https://doi.org/10.1364/OPTICA.505189>
- 3 Kwon, Y. et al. Computational conjugate adaptive optics microscopy for longitudinal through-skull imaging of cortical myelin. *Nature Communications* 14, 105 (2023). <https://doi.org/10.1038/s41467-022-35738-9>
- 4 Bureau, F. et al. Three-dimensional ultrasound matrix imaging. *Nature Communications* 14, 6793 (2023). <https://doi.org/10.1038/s41467-023-42338-8>

Reviewer #2 (Remarks to the Author):

I am convinced by the authors' response and the new results included in the paper (Fig.~8) and the Supplementary Material.

We thank the reviewer for their valuable feedback again.

Just one minor comment:

The authors refer to CLASS and the distortion matrix approach as conventional AO methods. This terminology can be misleading as those methods are computational and not conventional in the field of adaptive optics. They are both based on the post-processing of the reflection/transmission matrix, which is rather new in the field of adaptive optics. I would suggest to refer to those approaches as matrix imaging methods or alternative CAO methods (CAO: computational adaptive optics).

Thank you for pointing out this oversight. As you suggested, the manuscript has been updated accordingly.

Reviewer #3 (Remarks to the Author):

Reviewer #3 [conclusion]:

The concept is elegant, and the proof-of-concept experiments are convincing. But the applicability seems limited because some theoretical derivations aren't clear and because the limitations of the techniques aren't much discussed. Some additional points need to be clarified.

Reviewer #3 [Comment 3.1]

"At first, I'm a bit confused with the notations used by the authors. In equation (4), T is defined as an equivalent to the scattering matrix of the sample. But if T is a complex matrix, what allow the authors to go from a matrix product to an Hadamard product? I feel an assumption is missing here that will help the reader to understand the limitation of this approach."

The answer of "convolution theorem" make sense, but the formula in the reply to the review #3 is not correct.

1. We assume dr is supposed to be dr_1 ?

Yes, you are correct, it should be dr_1 . We have now provided the correct formula below.

2. The highlighted formula is not the convolution theorem, which says the Fourier transform of the convolution of two functions equals the product of their Fourier transforms.

We apologize for the mistake. The formula should be

$$\int P(r_1) *_r_1 T(r_1, r_2) e^{-ik_1 r_1} dr_1 = P(k_1) T(k_1, r_2),$$

$$P(k_1) = \int P(r_1) e^{-ik_1 r_1} dr_1,$$

$$T(k_1, r_2) = \int T(r_1, r_2) e^{-ik_1 r_1} dr_1,$$

where $*_{r_1}$ denotes convolution with respect to r_1 .

Reviewer #3 [Comment 3.2]:

"I feel there is a Δk range where the wavefronts gradients can be successfully reconstructed (whatever the angular memory effect of the sample). If Δk is too small, the sensitivity might be too low to measure the gradient in presence of noise. If it's too large the derivative might be misestimate. Can the authors precise/quantify the Δk range validity of their approach?"

The authors' reply merely re-iterates and elaborates the Reviewer's comment that "If Δk is too small, the sensitivity might be too low to measure the gradient in presence of noise. If it's too large the derivative might be misestimate."

The proper reply to the question is “From this observation, we conclude that the upper bound of the tilt angle is approximately the memory effect range ($\Delta k = 2\pi 0.022/\lambda$, third column in Fig. R9)”. Note that this is based on a single experimental measurement on a 100 μm thick human tissue. No systematic quantification was provided. Further, no evidence that this angular range is representative of a sample of general interest was provided.

Thank you for the clarification. The upper bound of the tilt angle is determined by the reduction in tilt/tilt correlation, which is predicted by the optical memory effect (Eq. 3 in the main text)^{5,6}.

The memory effect range can be estimated from optical parameters using the formula provided in reference⁶. For typical biological tissues with a transport mean free path of $l_t = 1 \text{ mm}$,⁷ the typical memory effect range for 100- μm thick tissues ($L = 100 \mu\text{m}$) can be estimated as $\sqrt{24l_t/L^3} = 0.16 \mu\text{m}^{-1}$.

Since this upper bound is much larger than the tilt angle used in the main text ($\Delta k = 2\pi 0.0054/\lambda = 0.064 \mu\text{m}^{-1}$), our method is applicable to biological samples with a thickness of 100 μm in general.

Reviewer #3 [Comment 3.3]

“To go further, this approach requires some angular memory effect. How much? The authors touched this issue with the 100 μm thick sample in figure 5. But it would be interesting to understand better. Can the authors link the memory effect range needed to the Δk range validity of their approach to estimate the tissue thickness they will be able to image?”

This comment is related to the Comment 3.2, as the authors pointed out. The authors did not provide additional measurements nor a calculation to directly address the reviewer’s question. The reply was based on the observation of one 50- μm thick tissue sample and one 100- μm thick tissue sample.

The criteria for “a correction of up to 50 μm thickness is practically sufficient” remains unclear.

As the reviewer suggested, the maximum thickness can be obtained by equating the memory effect range to the available tilt angle, which, based on the tilt angle used in the main text, results in an approximate maximum thickness of 180 μm .

$$\sqrt{24l_t/L_{\text{max}}^3} = 0.064 \mu\text{m}^{-1}$$

$$L_{\text{max}} = 180 \mu\text{m}$$

However, it is important to highlight that our method can extend beyond this thickness when combined with time-gating techniques, such as those used in Optical Coherence Tomography (OCT). In such cases (Supplementary Section 1), the effective thickness is determined by the axial resolution of the OCT system, typically on the order of tens of micrometers. Thus, in time-gated reflection imaging, the memory effect range does not constrain the applicability of our method, allowing it to be effective for thicker samples than initially calculated.

Reviewer #3 [Comment 3.4]

"To distinguish between incident and outgoing aberrations, the authors assumed that the incident aberrations are the same for different incident wave vectors. It's equivalent to assume that the incident aberrations are only due to the optical system. Is it always true? In a realistic scenario, where the aberrations to correct are induced by the sample itself, both incident and outgoing aberrations will depend on the incident wave vectors. Will the approach suggested by the authors still work?"

The authors clarified the theoretical derivation to address this comment. However, Reviewer #3 was also asking if this approach works for correcting sample-induced aberrations. For this case, the authors provide little experimental support.

To address the reviewer's concern, we had conducted an additional experiment in which two aberrating plastic pieces were inserted into both the incoming and outgoing beam paths. The result, presented in Fig. 8 of the revised manuscript, demonstrates the ability of our method to correct for aberrations in both directions.

Figure RR2: Sample with aberrating layers in both incoming and outgoing beam paths. An excerpt from Fig. 8 in the revised manuscript.

Reviewer 3 [Comment 3.5]

"Some phrasings are misleading. I quote from the Discussion and Conclusion: "It achieves effective aberration correction in samples up to 100- μ m thick", the authors didn't show these experiments. They only show an aberration measurement through a 100- μ m tissue with an indirect validation."

In Figure 6c the authors show an increase of 3D phase correlation after AO correction, yet the overall correlation is low (< 0.1), i.e. explaining $< 1\%$ of the variance. It is unclear what the criteria is for "achieving effective aberration correction" is in a practical sense.

We based our statement on the fact that the peak correlation is enhanced by more than tenfold. While we

acknowledge that a correlation (or Strehl ratio) of 0.1 may not be optimal in all cases, we believe it serves as a reasonable limit case in the context of our study. To provide clearer communication, we have removed the term "effective" from the text.

The remaining comments were for clarification and the authors have addressed them.

We thank the reviewer for the careful feedback.

References

- 5 Feng, S., Kane, C., Lee, P. A. & Stone, A. D. Correlations and Fluctuations of Coherent Wave Transmission through Disordered Media. *Physical Review Letters* 61, 834-837 (1988). <https://doi.org:10.1103/PhysRevLett.61.834>
- 6 Osnabrugge, G., Horstmeyer, R., Papadopoulos, I. N., Judkewitz, B. & Vellekoop, I. M. Generalized optical memory effect. *Optica* 4, 886-892 (2017). <https://doi.org:10.1364/OPTICA.4.000886>
- 7 Jacques, S. L. Optical properties of biological tissues: a review. *Physics in Medicine & Biology* 58, R37 (2013).

Reviewer #4 (Remarks to the Author):

We thank the reviewer for the review again.

In response - The authors have failed to apply their method to realistic tissue, which is the gold standard in this field.

We sincerely appreciate your valuable comments. We acknowledge that our previous response may not have fully addressed your concerns. Upon further review, we now understand your question pertains to whether our method can be applied under general biomedical imaging conditions where adaptive optics, as referenced in your citations, is typically utilized. The answer is affirmative.

To substantiate this, we applied our method to reflection imaging data of the human cornea at a depth of 105 μm , obtained from a publicly available dataset [Balondrade P. et al., Nat. Photon. 18, 1097–1104 (2024), doi.org/10.5281/zenodo.8407618]. Additionally, we have made the code used for this analysis available for your review (<https://github.com/BMOLKAIST/AberrationMatrix/tree/main/RMI>).

a Coherently compounded images

b Acquired aberrations

Figure A1 (S3): Aberration correction in opaque human cornea at a depth of 105- μm . **a**, Coherently compounded images. Left: uncorrected, middle: corrected using our method, right: corrected using CLASS. **b**, Aberrations obtained at 15×15 points using our method (left), and CLASS (right). Experimental data was obtained from doi.org/10.5281/zenodo.8407618.

Our method significantly enhances image quality compared to the uncorrected image. Although the CLASS method also achieves some improvement, our approach yields better contrast, particularly within the region highlighted by the orange circle in Fig. A1. This finding is especially noteworthy considering that the experimental setup was not specifically optimized for our technique; the minimum tilt angle in this dataset is substantially larger (more than ten times) than that employed in our manuscript.

However, performing a quantitative analysis in actual biomedical settings presents inherent challenges due to the absence of a ground truth image. Embedding a highly reflective resolution target within tissue remains unrealistic. Consequently, we used an artificial aberrating layer (polymer or opaque plastic pieces) to evaluate our method.

Additionally, we have added a section to the main text discussing simulations of reflection imaging through biological tissues under general imaging conditions. These simulations demonstrate a direct improvement in the Strehl ratio, further supporting the applicability of our method in biomedical imaging contexts.

It's unclear under what realistic biomedical imaging condition corresponds to an artificially introduced aberration element. It's also unclear why using "a coverslip coated with polymer" enables "systematic evaluation" of the technique. While employing polymer layers is used for testing the method, the practical value of the method should be evaluated using realistic biological samples. In exactly the studies cited above:

As demonstrated by the results presented, our method encounters no additional challenges when applied to deep tissue imaging beyond what is discussed in the manuscript. The materials we used to induce aberrations artificially—polymers (Figs. 4-7) and opaque plastic pieces (Fig. 8)—possess higher refractive indices than typical biological tissues, resulting in significantly stronger aberrations. This makes the correction process considerably more demanding. Nevertheless, our method consistently achieves better performance under these conditions.

In realistic biomedical imaging, aberrations predominantly arise at interfaces with significant refractive index mismatches, such as the tissue-medium or skull-brain boundaries. This observation underlies the principle of conjugate AO, which aims to correct aberrations at planes conjugate to these interfaces^{1,2}. Similarly, in the cornea sample, aberrations also stem from its curved surface. Based on this, we expect that placing polymer or plastic pieces can effectively simulate such interfaces.

In addition, the experiment shown in Fig. 8, which involves imaging through opaque plastic pieces, implies our ability to image through damaged coverslips, a relevant issue in high-resolution imaging. Plastic coverslips, especially when damaged, tend to have reduced optical clarity compared to standard glass coverslips. Our method effectively addresses these challenges, showcasing its robustness in real-world conditions where coverslip damage is a potential concern.

References

1. Park, J.-H., Sun, W. & Cui, M. High-resolution in vivo imaging of mouse brain through the intact skull. *Proc. Natl Acad. Sci. USA* **112**, 9236–9241 (2015).
2. Yoon, S., Lee, H., Hong, J.H. *et al.* Laser scanning reflection-matrix microscopy for aberration-free imaging through intact mouse skull. *Nat Commun* **11**, 5721 (2020). <https://doi.org/10.1038/s41467-020-19550-x>

By the same argument, the following similar quantification on the same image (yellow boxed) suggests that the variance (and coefficient of variation, CV) in the corrected tomogram is slightly smaller than that in the polymer-free tomogram.

We appreciate the opportunity to clarify this point further. We apologize for not providing a more detailed explanation regarding our use of variance as a metric for sharpness. Unlike conventional imaging techniques that rely on measuring light intensity, our tomographic reconstruction technique derives the refractive index from the phase of light. In this context, a higher variance does not necessarily indicate a clearer image. Aberrations, for instance, can introduce a background phase, resulting in refractive index variations across the entire field of view. As such, using refractive index variance across the full sample area is not a straightforward indicator of aberration correction efficacy.

Figure A2 (S9): Additional analysis of aberration correction on 50- μm -thick tissue. **a**, Zoomed views of the polymer-added (top), corrected (middle), and polymer-free (bottom) tomograms. 3D refractive index stacks shows two red blood cells in contact at the first plane, gradually separating as the axial position (Δz), relative to section #2 defined in Fig. 5, increases. **b**, Refractive index profiles along the selected lines (red dashed line in a). The corrected tomogram exhibits more than twice refractive index variation (gray arrow) as the gap between two RBCs widens, compared to the polymer-free tomogram. **c**, Refractive index images of RBCs at three depths, comparing the corrected (left) and polymer-free (right) tomograms. The profiles along the selected lines (right) indicate a higher RI contrast of in-focus RBCs in the corrected tomogram.

In our previous analysis, variance was used specifically to measure the sharpness of RBCs, which are easily identifiable and exhibit distinct refractive index boundaries. In this updated analysis, we provide a more detailed examination of RBC sharpness in the reconstructed refractive index tomograms. In Fig. A2a, we present zoomed views of two RBCs initially in contact at the first plane ($\Delta z = -2.1$ μm). As the RBCs separate with increasing axial coordinates (Δz), we analyzed the refractive index at the gap between them. As shown in Fig. A2b, the refractive index variation at this gap is more than twice as high in the corrected tomogram compared to the polymer-free tomogram, confirming enhanced resolution. Additionally, overall refractive index contrast is improved after correction, as illustrated in Fig. A2c, where in-focus RBCs at various depths exhibit higher contrast in the corrected tomograms.

Afterall, the improvement of the “corrected tomogram” over “polymer-free tomogram” appears to be very weak (if any), especially when compared to the demonstrations in the referenced studies above.

The analysis in Fig. A2 confirms that RBCs in the corrected tomograms exhibit sharper edges than those in the polymer-free tomogram. As noted, aberration levels are generally weak in tissues around 100 μm thick, excluding surface light deflection (left column of Fig. 9). However, as in the corneal imaging, the structure of the interface may generate considerable aberration. We aimed to mimic this effect using polymer and plastic pieces, which induce even stronger aberrations than those typically encountered in realistic corneal imaging, as shown in Fig. A1b.

Based on the provided data, we believe our method has proven applicability across a wide range of realistic imaging scenarios. Our demonstration emphasizes the optical principle of adaptive optics (memory effect) in biological samples, a topic of interest not only for biomedical researchers but also for physicists studying light transport. Additionally, our work creates new opportunities to extend current adaptive optics techniques to more

challenging environments. Importantly, we also address sample dynamics, a well-known and significant obstacle in this field.

Reviewer #3 [conclusion]:

The concept is elegant, and the proof-of-concept experiments are convincing. But the applicability seems limited because some theoretical derivations aren't clear and because the limitations of the techniques aren't much discussed. Some additional points need to be clarified.

Reviewer #3 [Comment 3.1]

“At first, I’m a bit confused with the notations used by the authors. In equation (4), T is defined as an equivalent to the scattering matrix of the sample. But if T is a complex matrix, what allow the authors to go from a matrix product to an Hadamard product? I feel an assumption is missing here that will help the reader to understand the limitation of this approach.”

The answer of “convolution theorem” make sense, but the formula in the reply to the review #3 is not correct.

This is based on the convolution theorem, $\int P(r_1) * T(r_1, r_2) dr = P(k_1)T(k_1, r_2)$, which states that the convolution operation $[P(r_1) *]$ becomes an element-wise multiplication in the k -space representation. Therefore, the effect of the aberration can be described by the Hadamard product by changing the basis from the

1. We assume dr is supposed to be dr_1 ?
2. The highlighted formula is not the convolution theorem, which says the **Fourier transform of** the convolution of two functions equals the product of their Fourier transforms.

Reviewer #3 [Comment 3.2]:

“I feel there is a Δk range where the wavefronts gradients can be successfully reconstructed (whatever the angular memory effect of the sample). If Δk is too small, the sensitivity might be too low to measure the gradient in presence of noise. If it's too large the derivative might be misestimate. Can the authors precise/quantify the Δk range validity of their approach?”

The authors' reply merely re-iterates and elaborates the Reviewer's comment that *“If Δk is too small, the sensitivity might be too low to measure the gradient in presence of noise. If it's too large the derivative might be misestimate.”*

The proper reply to the question is *“From this observation, we conclude that the upper bound of the tilt angle is approximately the memory effect range ($\Delta k = 2\pi \cdot 0.022/\lambda$, third column in Fig. R9)”. Note that this is based on a single experimental measurement on a 100 μm thick human tissue. No systematic quantification was provided. Further, no evidence that this angular range is representative of a sample of general interest was provided.*

Reviewer #3 [Comment 3.3]

“To go further, this approach requires some angular memory effect. How much? The authors touched this issue with the 100 μm thick sample in figure 5. But it would be interesting to understand better. Can the authors link the memory effect range needed to the Δk range validity of their approach to estimate the tissue thickness they will be able to image?”

This comment is related to the Comment 3.2, as the authors pointed out. The authors did not provide additional measurements nor a calculation to directly address the reviewer's question. The reply was based on the observation of one 50- μm thick tissue sample and one 100- μm thick tissue sample.

The criteria for “a correction of up to 50 μm thickness is practically sufficient” remains unclear.

Reviewer #3 [Comment 3.4]

“To distinguish between incident and outgoing aberrations, the authors assumed that the incident aberrations are the same for different incident wave vectors. It’s equivalent to assume that the incident aberrations are only due to the optical system. Is it always true? In a realistic scenario, where the aberrations to correct are induced by the sample itself, both incident and outgoing aberrations will depend on the incident wave vectors. Will the approach suggested by the authors still work?”

The authors clarified the theoretical derivation to address this comment. However, Reviewer #3 was also asking if this approach works for correcting sample-induced aberrations. For this case, the authors provide little experimental support.

Reviewer 3 [Comment 3.5]

“Some phrasings are misleading. I quote from the Discussion and Conclusion: “It achieves effective aberration correction in samples up to 100- μm thick”, the authors didn’t show these experiments. They only show an aberration measurement through a 100- μm tissue with an indirect validation.”

In Figure 6c the authors show an increase of 3D phase correlation after AO correction, yet the overall correlation is low (< 0.1), i.e. explaining $< 1\%$ of the variance. It is unclear what the criteria is for “achieving effective aberration correction” is in a practical sense.

The remaining comments were for clarification and the authors have addressed them.

In response - The authors have failed to apply their method to realistic tissue, which is the gold standard in this field.

“We appreciate the reviewer's concern regarding the practical applicability of our method. The use of artificial aberrations in our experiments was a deliberate choice to validate the robustness and flexibility of our approach under controlled yet realistic conditions. This is a widely accepted practice in adaptive optics, as it enables systematic evaluation and comparison of novel techniques.”

It's unclear under what realistic biomedical imaging condition corresponds to an artificially introduced aberration element. It's also unclear why using “a coverslip coated with polymer” enables “systematic evaluation” of the technique.

“While we acknowledge that more explicit real-world examples could further provide additional clarity, our goal was to maintain the broad applicability of the method by avoiding assumptions about specific forms of aberrations or sample types. This approach ensures that our method is generalizable and relevant across a variety of commonly encountered imaging contexts, such as reflection imaging of biological specimens. Additionally, introducing artificial aberrations for validation has been well-established in the field as a means of method validation. For instance, techniques such as CLASS^{1,2} employed polymer layers as artificial aberrating layers, and similar approaches are widely accepted for testing novel methods.”

While employing polymer layers is used for testing the method, the practical value of the method should be evaluated using realistic biological samples. In exactly the studies cited above:

- Reference 1 Kang et al 2017 Nature Communications:
 - Fig 4 demonstrates aberration correction for a target underneath a rat brain tissue and shows a clear improvement in the image quality after correction.
 - Fig 5 demonstrates imaging the hyphae of *Aspergillus* cell in a rabbit's cornea and shows an improvement in imaging quality after aberration correction.
- Reference 2 Moon et al 2024 Optica:
 - Fig 5 demonstrates SHG imaging of a zebrafish muscle with clear improvement after aberration correction.

“We appreciate the reviewer's emphasis on the importance of systematic quantification. To address this, we have incorporated a more rigorous quantitative analysis of the improvement over the polymer-free image by introducing the variance of the refractive index as a sharpness metric. This metric, shown in the revised Fig. 5f, provides a reliable means of evaluating image clarity, as a larger variance typically correlates with reduced blurring and enhanced detail in the image.”

Specifically, the corrected tomogram shows a 3.7-fold increase in variance when compared to the polymer-added tomogram, and a 1.3-fold increase compared to the polymer-free tomogram. These results clearly demonstrate the significant improvement in image sharpness achieved through aberration correction, particularly in the vicinity of the red blood cells. This systematic quantification confirms the enhanced axial and overall image quality, thereby validating the effectiveness of our method.”

By the same argument, the following similar quantification on the same image (yellow-boxed) suggests that the variance (and coefficient of variation, CV) in the corrected tomogram is slightly smaller than that in the polymer-free tomogram. After all, the improvement of the “corrected tomogram” over “polymer-free tomogram” appears to be very weak (if any), especially when compared to the demonstrations in the referenced studies above.

ImageJ quantification using “Measure” (ctrl + m): Mean ± StdDev (CV)

In the yellow-boxed region: 90.6 ± 8.8 (9.7%)

In the yellow-boxed region: 92.0 ± 12.8 (13.9%)

In the yellow-boxed region: 93.9 ± 13.3 (14.2%)

The value of mean and variance were the same for both the RGB image and grayscale 16-bit image.

In response - The authors have failed to apply their method to realistic tissue, which is the gold standard in this field.

“We appreciate the reviewer's concern regarding the practical applicability of our method. The use of artificial aberrations in our experiments was a deliberate choice to validate the robustness and flexibility of our approach under controlled yet realistic conditions. This is a widely accepted practice in adaptive optics, as it enables systematic evaluation and comparison of novel techniques.”

It's unclear under what realistic biomedical imaging condition corresponds to an artificially introduced aberration element. It's also unclear why using “a coverslip coated with polymer” enables “systematic evaluation” of the technique.

“While we acknowledge that more explicit real-world examples could further provide additional clarity, our goal was to maintain the broad applicability of the method by avoiding assumptions about specific forms of aberrations or sample types. This approach ensures that our method is generalizable and relevant across a variety of commonly encountered imaging contexts, such as reflection imaging of biological specimens. Additionally, introducing artificial aberrations for validation has been well-established in the field as a means of method validation. For instance, techniques such as CLASS1,2 employed polymer layers as artificial aberrating layers, and similar approaches are widely accepted for testing novel methods.”

While employing polymer layers is used for testing the method, the practical value of the method should be evaluated using realistic biological samples. In exactly the studies cited above:

- Reference 1 Kang et al 2017 Nature Communications:
 - Fig 4 demonstrates aberration correction for a target underneath a rat brain tissue and shows a clear improvement in the image quality after correction.
 - Fig 5 demonstrates imaging the hyphae of Aspergillus cell in a rabbit's cornea and shows an improvement in imaging quality after aberration correction.
- Reference 2 Moon et al 2024 Optica:
 - Fig 5 demonstrates SHG imaging of a zebrafish muscle with clear improvement after aberration correction.

“We appreciate the reviewer's emphasis on the importance of systematic quantification. To address this, we have incorporated a more rigorous quantitative analysis of the improvement over the polymer-free image by introducing the variance of the refractive index as a sharpness metric. This metric, shown in the revised Fig. 5f, provides a reliable means of evaluating image clarity, as a larger variance typically correlates with reduced blurring and enhanced detail in the image.”

Specifically, the corrected tomogram shows a 3.7-fold increase in variance when compared to the polymer-added tomogram, and a 1.3-fold increase compared to the polymer-free tomogram. These results clearly demonstrate the significant improvement in image sharpness achieved through aberration correction, particularly in the vicinity of the red blood cells. This systematic quantification confirms the enhanced axial and overall image quality, thereby validating the effectiveness of our method.”

By the same argument, the following similar quantification on the same image (yellow-boxed) suggests that the variance (and coefficient of variation, CV) in the corrected tomogram is slightly smaller than that in the polymer-free tomogram. After all, the improvement of the “corrected tomogram” over “polymer-free tomogram” appears to be very weak (if any), especially when compared to the demonstrations in the referenced studies above.

ImageJ quantification using “Measure” (ctrl + m): Mean ± StdDev (CV)

In the yellow-boxed region: 90.6 ± 8.8 (9.7%)

In the yellow-boxed region: 92.0 ± 12.8 (13.9%)

In the yellow-boxed region: 93.9 ± 13.3 (14.2%)

The value of mean and variance were the same for both the RGB image and grayscale 16-bit image.